# Self-activated superhydrophilic green ZnIn₂S₄ realizing solar-driven overall water splitting: close-to-unity stability for a full daytime

Wei-Kean Chong[1], Boon-Junn Ng[1], Yong Jieh Lee[1], Lling-Lling Tan[1], Lutfi Kurnianditia Putri[1], Jingxiang Low[1,2], Abdul Rahman Mohamed[3] & Siang-Piao Chai [1] ✉

Engineering an efficient semiconductor to sustainably produce green hydrogen via solar-driven water splitting is one of the cutting-edge strategies for carbon-neutral energy ecosystem. Herein, a superhydrophilic green hollow ZnIn₂S₄ (gZIS) was fabricated to realize unassisted photocatalytic overall water splitting. The hollow hierarchical framework benefits exposure of intrinsically active facets and activates inert basal planes. The superhydrophilic nature of gZIS promotes intense surface water molecule interactions. The presence of vacancies within gZIS facilitates photon energy utilization and charge transfer. Systematic theoretical computations signify the defect-induced charge redistribution of gZIS enhancing water activation and reducing surface kinetic barriers. Ultimately, the gZIS could drive photocatalytic pure water splitting by retaining close-to-unity stability for a full daytime reaction with performance comparable to other complex sulfide-based materials. This work reports a self-activated, single-component cocatalyst-free gZIS with great exploration value, potentially providing a state-of-the-art design and innovative aperture for efficient solar-driven hydrogen production to achieve carbon-neutrality.

Hydrogen (H₂), the smallest molecule with the largest specific energy content, is an emerging alternative to the traditional fossil energy for achieving carbon neutrality. Inspired by the natural photosynthesis, utilization of semiconductors to drive photocatalytic water splitting for clean H₂ and oxygen (O₂) production has been widely explored[1,2]. Photocatalytic water splitting could serve as an important step towards a more sustainable energy future, as it allows the conversion of abundant photon energy into chemical energy as well as the storage of solar energy in the form of high energy-content H₂. Within this framework, the selection of an appropriate photocatalyst is extremely critical as it directly governs the efficiency of the solar-driven H₂ generation. Lately, metal chalcogenide semiconductors have gained enormous attention in this field owing to the favorable visible-light response ability, large abundancy, and diversified chemical structures[3,4].

In particular, two-dimensional (2D) hexagonal ZnIn₂S₄ (ZIS), a well-known ternary metal chalcogenide, has garnered significant attention in photocatalytic water splitting. The appealing features of ZIS such as tunable band gap, high photosensitivity towards visible light and favorable conduction band potential with a strong reduction

[1]Multidisciplinary Platform of Advanced Engineering, Department of Chemical Engineering, School of Engineering, Monash University Malaysia, Jalan Lagoon Selatan, 47500 Bandar Sunway, Selangor, Malaysia. [2]Department of Applied Chemistry, University of Science and Technology of China (USTC), 96 Jinzhai Road, Hefei, Anhui 230026, PR China. [3]School of Chemical Engineering, Universiti Sains Malaysia, 14300 Nibong Tebal Pulau Pinang, Malaysia. ✉e-mail: chai.siang.piao@monash.edu

capability making it a prominent candidate for $H_2$ evolution reaction (HER)[5,6]. Different morphologies of ZIS have been developed till date, with dominance of microspheres and nanosheets attributed to the ease of fabrication, structural stability and diverse application[7]. Nonetheless, microspherical ZIS suffers from low exposure of active surface area due to the bilayer self-assembly into a three-dimensional (3D) microsphere, covering some active sites in the core. Despite 2D nanosheet ZIS possessing more surface area per unit volume, it suffers from severe aggregation and interlayer stacking with surrounding nanosheets to modulate its high surface energy[8,9]. This leads to reduction of available active surface sites and worsened surface charge separation, deteriorating the photocatalytic performance. Thus, it is critical to alter the morphology of ZIS to minimize the rate of charge carrier recombination, while simultaneously possessing high availability of active surface area for the photocatalytic reaction. Furthermore, the self-oxidation of ZIS and deficiency of catalytic $O_2$ evolution reaction (OER) sites serve as long hidden bottleneck issues that limit the application of ZIS in overall water splitting[10]. Several strategies could be employed in addressing the aforesaid criteria. Primarily, the structure of ZIS could be designed to enhance light scattering for improved light utilization and to prevent interlayer aggregation. Secondly, the introduction of S vacancy ($S_v$) into the framework to suppress electron-hole pair recombination, promote charge redistribution and facilitate photoreaction[11,12]. Following that, the density of the active edge S atoms in (110) facet should also be increased via selective unleashing of the edge sites to favor the photogeneration of $H_2$[13,14]. Furthermore, the surface hydrophilicity could be improved to facilitate water molecule interactions to drive a more vigorous water-splitting reaction. Concomitantly, high surface wettability could promote efficient mass transfer of water molecules to active surface and expedite instantaneous release of generated gas bubble to maintain ubiquitous availability of active sites[15–18].

Herein, a distinctive superhydrophilic green $ZnIn_2S_4$ (gZIS) was constructed in this work via a one-step in-situ solvothermal synthetic route. The gZIS with hollow hierarchical framework is found to possess higher specific surface area with more exposed active facets. The superhydrophilic surface enhances interaction with surrounding water molecules to drive water decomposition. Besides, gZIS experiences an optical absorption property analogous to natural leaves, utilizing both the high and low wavelength of solar light to generate electron-hole pairs for photoreaction. The defects within the structure further regulate the charge redistribution and activate the inert basal plane with facile charge transfer and enhanced surface reaction. The first-principle calculations provide theoretical insights and verify the significant roles of vacancies in electronic properties modulation, water molecules interaction, HER and OER surface kinetics improvement. As a result, this self-activated gZIS demonstrated its capability in catalyzing solar-driven overall water splitting with close-to-unity stability for a full daytime reaction. In addition, the single-component cocatalyst-free gZIS exhibited an apparent quantum yield (AQY) and solar-to-hydrogen conversion efficiency (STH) that is comparable to other noble-metal loaded and complex sulfide-based photocatalysts. These groundbreaking deliveries represent a significant breakthrough in addressing the longstanding concealed obstacles of sulfide-based materials, particularly the unassisted overall water splitting capability and photostability. This discovery will pave a way towards the development of high-performing photocatalysts to achieve efficient and sustainable overall water splitting without the incorporation of expensive noble metal cocatalysts.

## Results and discussions
### Morphological design fundamentals and structural characterization
A conventional yellow 3D microspherical ZIS was synthesized via hydrothermal route as shown in Fig. 1a. In detail, $H_2O$ molecules

donate lone pair electrons to $Zn^{2+}$ via dative bonding. Zn-aquo complex is then formed, followed by the transformation into tetrahedral $[Zn(TAA)_4]^{2+}$ complex in the solution upon ligand exchange. Meanwhile, the formation of two distinctive aquo complexes of $In^{3+}$ give rise to In-TAA complexes with varying coordination number, i.e., tetrahedral $[In(TAA)_4]^{3+}$ and octahedral $[In(TAA)_6]^{3+}$[19,20]. Subsequently, three metal sulfide species ($Zn-S_4$, $In-S_4$, and $In-S_6$) are formed under the hydrothermal condition, which spontaneously combine into 2D hexagonal bilayer ZIS nanosheets in S-Zn-S-In-S-In-S stacking sequence. Due to the high surface energy of 2D layered structure, the ZIS nanosheets would self-assemble into 3D microspheres to moderate the surface energy for a more thermodynamically stable form. On the other hand, a unique green gZIS was successfully fabricated by employing ethylene glycol (EG) as the solvent. In the presence of EG, $Zn^{2+}$ and $In^{3+}$ are subjected to the formation of metal glycol complexes with larger radii than the respective metal aquo complexes[21]. During the solvothermal reaction, the complexes interact with $S^{2-}$ to form different metal sulfide species that lead to nanolayer construction and subsequently self-assemble into unique gZIS framework. In this context, EG serves pivotal roles in the formation of the final structure: (i) Firstly, it acts as a surfactant that moderates the surface tension at the boundaries between particles, lowering particle aggregation and promoting anisotropic assembly into hollow hierarchical cavity-network configurations[19,22], (ii) it also reduces the valence state of $Zn^{2+}$ and $In^{3+}$ leading to a decrease in coordination abilities with $S^{2-}$[21,23,24], and (iii) lastly, it reacts with the exposed S on the terminated surface inducing surface S removal, which collectively incur $S_v$ into the gZIS structure[25,26].

The morphologies and microstructures of the as-synthesized ZIS and gZIS were analyzed by field emission scanning electron microscopy (FESEM). As displayed in Fig. 1b, d and Supplementary Fig. 1, ZIS exhibits a basic configuration composing of closely assembled nanosheets flowerlike structure with densely packed core, which would deteriorate the exposure and utilization of active surface sites. In the absence of any surfactant, the morphology of ZIS is highly irregular and consists of micro-sized spheres varying from 3 to 10 μm. Conversely, EG-assisted solvothermal reaction is more conducive towards generating hollow gZIS lamellar framework from the loose interwoven of nanoflakes (see Fig. 1c, e and Supplementary Fig. 1). The numerous sparsely intersecting nanosheets not only exposes more expansive active surface sites for the photoredox reactions, but also ameliorates the multilevel reflection and scattering of incident photons for the enhancement of light absorption[27]. Ascribed to the presence of EG, the growth of gZIS is more uniform with equivalent diameters ranging from 1 to 2 μm. This would lead to a more homogenous distribution of active sites for photocatalytic reactions. The elemental compositions of ZIS and gZIS were investigated using energy-dispersive X-ray (EDX) spectroscopy. The EDX spectrum of pristine ZIS in Fig. 1f clearly reveals the even and spherical distribution of Zn, In, and S elements with Zn:In:S atomic ratio of 1.07:2.00:3.99, which is close to the ideal stoichiometric ratio of 1:2:4 (see Supplementary Table 1). The EDX mapping also confirmed the uniform coexistence of Zn, In, and S in the gZIS structure, with an S defective atomic ratio (see Fig. 1g and Supplementary Table 1). Citing to the normal levels of In atoms, the $S_v$ concentration of gZIS could be determined by comparing the S:In ratio of gZIS to that in pristine ZIS[28]. The EDX analysis reflects the presence of *ca.* 3% $S_v$ across the gZIS hierarchical framework as depicted in the EG-assisted synthesis reaction. The $S_v$ in the framework could act as electron traps to facilitate vectorial transport of photogenerated electrons as well as induce favorable lattice defects for charge redistribution and electronic properties modification[29–31].

Transmission electron microscopy (TEM) and high-resolution TEM (HRTEM) were employed to investigate the microcosmic characteristics of ZIS and gZIS. In concordance with the observation in

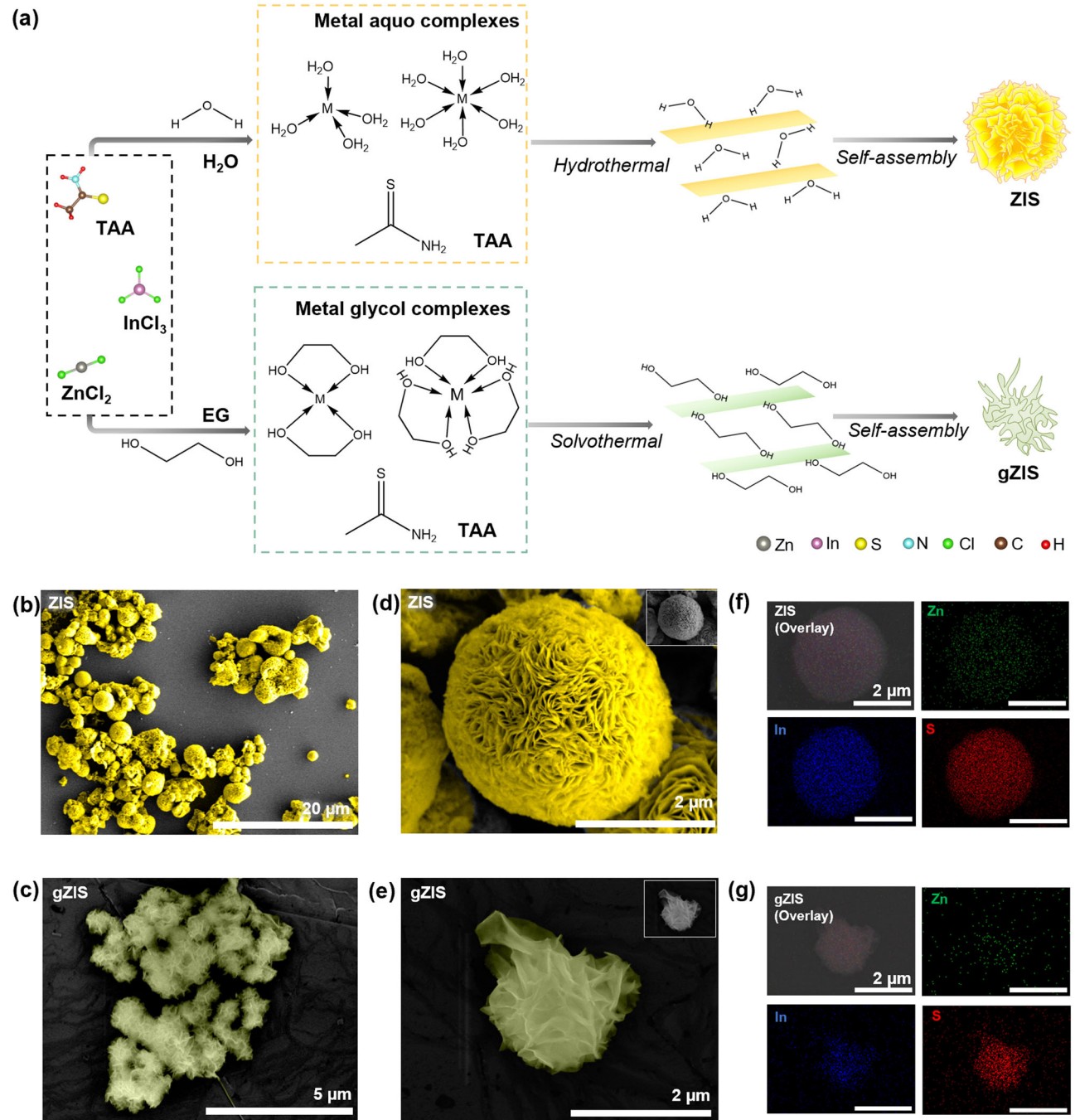

**Fig. 1 | Catalyst synthesis and morphological characterizations. a** Schematic of the formation of ZIS and gZIS. The charges of the complexes are omitted in the figure for clarity. M denotes the metal ions, either $Zn^{2+}$ or $In^{3+}$, present in the solution. False-colored FESEM images for (**b**) ZIS and (**c**) gZIS. Magnified false-colored FESEM view for (**d**) ZIS and (**e**) gZIS, with the insets showing the original FESEM images. EDX elemental mappings for (**f**) ZIS and (**g**) gZIS.

FESEM, TEM image of ZIS manifests the packed layer-by-layer assembly of nanosheets in constructing a large microstructure with a dense core (see Fig. 2a). Antithetical to the packed solid core ZIS structure, there are loose interwoven thin nanosheets in the smaller hollow hierarchical gZIS configurations (Fig. 2b). The 3D flimsy layer-to-layer interconnection of thin nanosheets of gZIS is advantageous in exposing more active sites for reactant species interaction, and at the same time preventing the undesired stacking and aggregation as-of observed in the standalone single phase nanosheet system[32]. Comparatively, the migration distance of photogenerated charge carriers could also be shortened in the smaller gZIS particle, favoring the participation of the electron-hole pairs in the photocatalytic reactions[33]. The HRTEM image in Fig. 2c presents the characteristic d-spacings of 0.32 and 0.19 nm, corresponding to the (102) and (110) of ZIS, respectively. There is nearly imperceptible distortion observed along the lattices of ZIS, signifying the successful construction of a pure pristine ZIS crystal. Directing attention to Fig. 2d, the lattice spacing of gZIS is *ca.* 0.19 nm which is assigned to the (110) plane. Unlike the pristine ZIS, lattice fringe distortions and defects are noticeable in the structure of gZIS owing to the presence of $S_v$. The asymmetrical distortions and defects could potentially induce instantaneous dipole moment which enhance the charge separation efficiency[34]. To shed light onto the structural atomic insights, first-principle density functional theory (DFT) was utilized to replicate the theoretical models for ZIS ($ZIS_T$) and gZIS

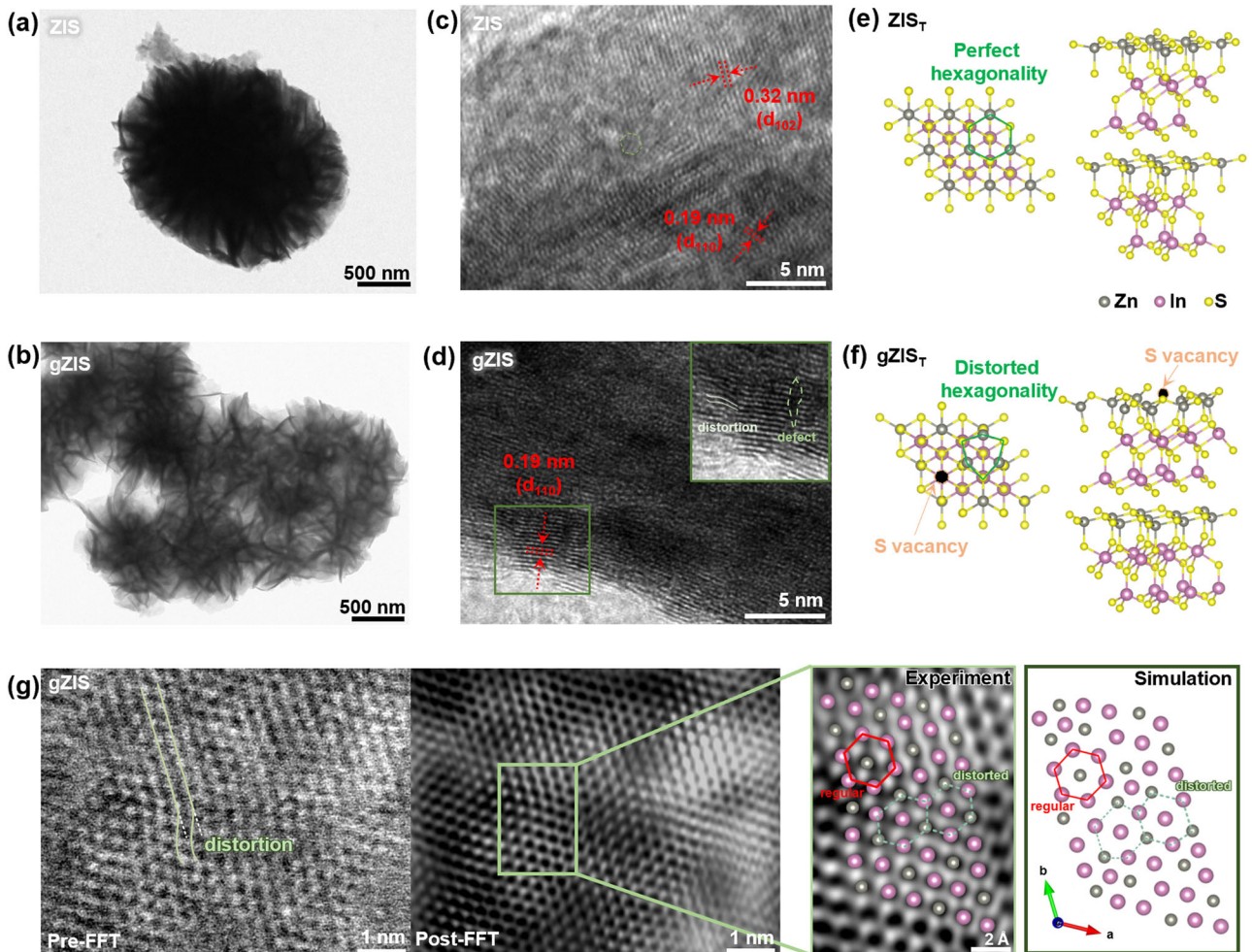

**Fig. 2 | Structural characterizations and analysis.** TEM images for (**a**) ZIS and (**b**) gZIS. HRTEM images for (**c**) ZIS and (**d**) gZIS, with an inset showing the enlarged region with lattice distortion and defects in gZIS. Theoretical structural models for (**e**) pristine ZIS$_T$ and (**f**) S-vacant gZIS$_T$. **g** Atomic-resolution spherical aberration- corrected BF-STEM imaging of gZIS with pre- and post-FFT. The magnified view shows the atomic arrangement with distorted hexagonal in concordance to the simulated result.

(gZIS$_T$). As demonstrated in Supplementary Fig. 2, different S$_v$ positions were introduced into the ZIS$_T$, with the formation favorability evaluated by the magnitude of formation energy (E$_{form}$). Removal of S from the (001) basal plane position (SV1) leads to a lower E$_{form}$, denoting a more energetically favorable structure of sulfur-vacant gZIS$_T$. Pristine ZIS$_T$ resembles a perfect hexagonal bilayer structure without any bond dislocation (Fig. 2e and Supplementary Fig. 3a–c). In contrast, the presence of Sv in the framework of gZIST alters the arrangement of neighboring atoms as illustrated in Fig. 2f and Supplementary Fig. 3d–f. The occurrence of theoretical lattice distortion is coherent with the experimental HRTEM observations that eventually distorts the endogenous hexagonality. In pursuit of disclosing further insights onto the structural information of gZIS, spherical aberration-corrected bright field scanning TEM (BF-STEM) was employed to provide atomic arrangement information of the framework as eluci-dated in Fig. 2g. Congruently, an obvious distortion was observed along the atomic alignment of gZIS attributed to the presence of S$_v$ which altered the intrinsic arrangement. Fast Fourier transformation (FFT) was then conducted to provide an amplified resolution of the atomic imaging. In point of fact, the relatively smaller S atoms, over-laying on top or underneath the Zn and In atoms (see Supplementary Fig. 4a), could be hardly visible under microscopic vision. Thus, the visibility of S atoms in the modeled structure is toggled-off in Fig. 2g (right-most) and Supplementary Fig. 4b for equitably juxtaposition.

The magnified post-FFT atomic imaging convincingly manifests the presence of distortion within the gZIS structure, which is in agreement with the simulated result. On one hand, the In-In hexagonal ring remains high regularity, implying the undisturbed arrangement of In atoms across the structure owing to the location of S$_v$ far from the In hosts. On another hand, two consecutive alternating Zn-In hexagonal rings experience slight deformation, signifying the presence of S$_v$ close to Zn atom which eventually induces disorderness around Zn hosts. The observations in the experimental findings align with the theore-tical simulations, which verified the existence of S$_v$ within gZIS struc-ture and affirm the location of S defect close to Zn atom along the basal plane. The existence of S$_v$ inevitably induces structural defects in gZIS framework which eventually lead to a charge redistribution.

## Surface electronic and physiochemical characteristics

The surface chemical states and elemental compositions of ZIS and gZIS were analyzed by X-ray photoelectron spectroscopy (XPS). As evident from the full survey scan spectrum in Supplementary Fig. 5, the presence of Zn, In and S peaks clearly delineate the coexistence of these elements in both the ZIS and gZIS sample, which is consistent with that in EDX analysis. Accentuated from Fig. 3a, the Zn 2$p$ metallic peak in ZIS splits into two individual peaks of 2$p_{3/2}$ (1021.38 eV) and 2$p_{1/2}$ (1044.38 eV). The Zn 2$p$ peaks of gZIS moderately upshift to higher binding energy values due to the loss of electrons and the

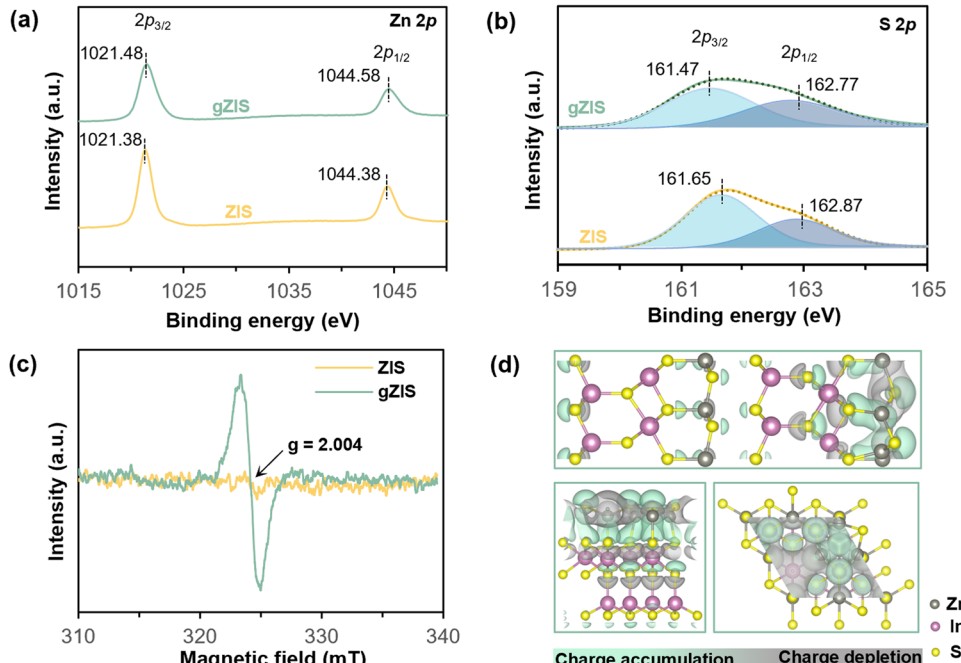

**Fig. 3 | Surface chemical and charge properties.** High-resolution XPS spectra of (**a**) Zn 2*p* and (**b**) S 2*p* for the as-synthesized samples. **c** EPR spectra for ZIS and gZIS indicating the presence of $S_v$. **d** Computed 3D charge density difference for gZIS$_T$, with the top showing the whole bilayer structure and the bottoms focus on the monolayer where $S_v$ is present. Gray and green areas dictate the charge depletion and accumulation isosurfaces, respectively.

reduction in coordination number of Zn to maintain the charge equilibrium with surrounding S elements[35,36], suggesting the presence of $S_v$ around Zn atom. Nonetheless, the spin-orbit splitting of S 2*p* confers bi-deconvoluted peaks in ZIS sample, namely 2$p_{3/2}$ (161.65 eV) and 2$p_{1/2}$ (162.87 eV) as shown in Fig. 3b. The intensity of the S 2*p* characteristic peaks in gZIS is patently weaker than that of ZIS, clearly demonstrating a lesser S content in gZIS. The XPS elemental composition suggests a consistent $S_v$ percentage in gZIS as previously observed in the EDX analysis (Supplementary Fig. 6). The presence of $S_v$ is further testified by the electron paramagnetic resonance (EPR) in Fig. 3c. The gZIS sample displays a sharp signal at a g-factor of *ca.* 2.004, which is not evident in pristine ZIS due to existence of $S_v$ only in the gZIS framework[37]. In addition, the two S 2*p* peaks of gZIS in Fig. 3b experience negative shift to lower binding energy, signifying the enrichment of electron cloud density around the S atoms[38,39]. The higher electronegativity of S contributes to a better tendency to attract electrons during the charge redistribution brought by $S_v$. DFT was then utilized to investigate the effect of $S_v$ on the charge distribution. As elucidated in the charge density difference from Fig. 3d, there is a noticeable charge redistribution in the gZIS$_T$ framework, with gray area showing the electron depletion zone and green area marking the electron accumulation region. It is not astonishing to observe the electron depletion zone in the $S_v$ location due to the loss of S atom. Coherent with the XPS finding, there is also a visible electron depletion around the Zn atom near to the $S_v$ along the basal plane. Additionally, the electron density not only increases in the intrinsically active S atoms at the (110) surface, but also gathers along the inherently unreactive S atoms at the basal plane. This defect-induced favorable charge redistribution could activate the inert basal plane for photoreactions as well as further boost the efficiency of $H_2$ production at the intrinsic active sites.

An essential aspect of assessing the catalytic performance is a thorough analysis of the crystal characteristics and surface properties of the samples. Therefore, X-ray diffraction (XRD) is utilized to acquire the crystal features. Harmonious with ZIS as presented in Fig. 4a, gZIS still preserves the two main hexagonal peaks in the structure, which are assigned to the (102) and (110) planes following PDF #065-2023[40]. The XRD spectra exhibit a conspicuous absence of any impurities peak, signifying the attainment of pure phases of ZIS and gZIS. It is widely recognized that the (110) facet of ZIS represents the most conducive site for HER, thus greater exposure of the (110) plane is desirable. Fascinatingly, the (110)-to-(102) peak intensity ratio increases from 0.92 in ZIS to 1.11 in gZIS. The greater-than-unity peak ratio clearly exhibits that (110) plane attains dominance in gZIS, providing higher exposure of HER active site. Following that, the Brunauer-Emmett-Teller (BET) specific surface area and pore size distribution of the samples were evaluated by the nitrogen adsorption-desorption isotherms. As manifested from Fig. 4b, pristine ZIS displays a conventional type IV isotherm with a calculated BET surface area of 71.83 m²·g⁻¹. Besides, the isotherm elucidates a large lag of $H_3$ hysteresis loop in the relative pressure ranged from 0.5 to 1.0, indicating the presence of wide distribution of non-uniformly shaped mesopores in the structure[41]. Such observation is consistent with the morphology previously observed in Fig. 1b, as well as the broad Barrett–Joyner–Halenda (BJH) pore size distribution in the inset of Fig. 4b. Conversely, the presence of EG facilitates even construction of gZIS with a relatively larger exposure of surface area (117.04 m²·g⁻¹) from its hollow hierarchical framework and a comparatively more uniform pore distribution concentrating at the smaller pore size. The well-dispersed gZIS exhibits higher specific surface area, which in turn offers more active sites for photocatalytic reaction. The hydrophilicity of the photocatalyst surface was also examined via the static water contact angle measurements (Fig. 4c). Pristine ZIS is found to be inherently hydrophilic with a contact angle of 68.6°, while gZIS transforms into superhydrophilic nature with a contact angle as low as 8.1°. The observed enhancement in the surface wettability of gZIS may be attributed to the higher degree of surface area exposure with increased surface roughness resulting from the dense small mesopores distributed throughout the large surface area of hollow framework. Furthermore, water dispersion test was conducted on ZIS and gZIS as shown in Supplementary Fig. 7. Both the samples were homogenously dissolved in DI water at similar concentrations. After

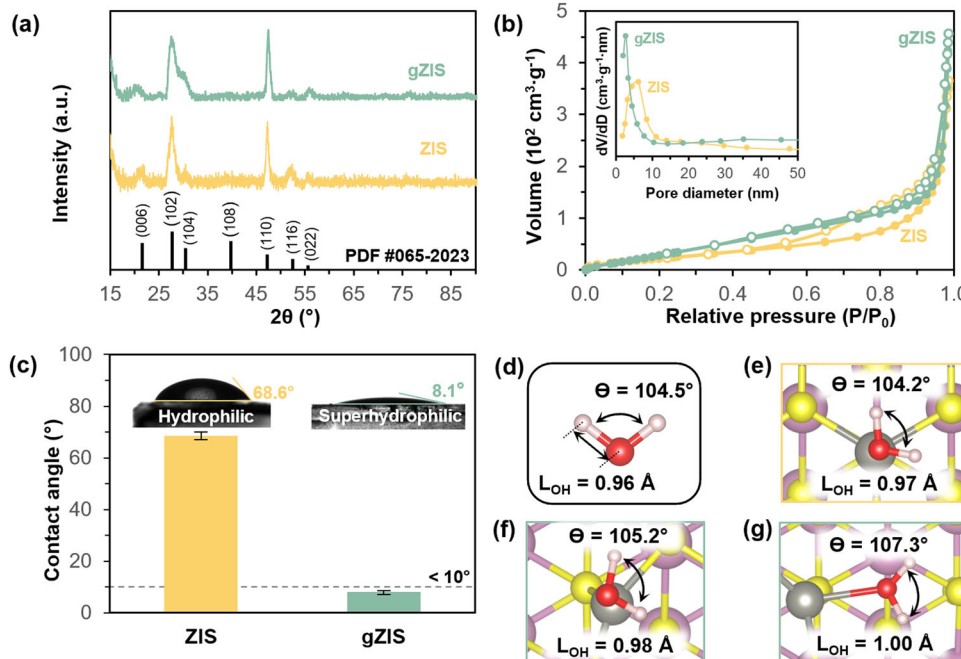

**Fig. 4 | Physical properties and water interaction study. a** XRD spectra for ZIS and gZIS. **b** Nitrogen adsorption-desorption isotherms of ZIS and gZIS with inset showing the respective pore size distribution. **c** Surface wettability static contact angle measurements for ZIS and gZIS; error bars represent the standard deviation from three independent runs. **d** Free water molecule with its respective O-H bond length and H-O-H bond angle. Theoretical modeling of water adsorption along the basal plane: (**e**) on Zn atom of $ZIS_T$, (**f**) on Zn atom of $gZIS_T$, and (**g**) in $S_v$ position of $gZIS_T$.

stewing for 5 min, there is observable precipitation of ZIS sedimented at the bottom, while gZIS remains well-dispersed in the solution. Some pale green precipitation of gZIS becomes discernible only after an elapsed time of 30 min. The increase in hydrophilicity and dispersion of gZIS would benefit the water adsorption capability and enhance the interface contact[42]. On top of the physical structural modification of gZIS promoting water engagement, the effect of $S_v$ onto the water interaction was also explored from DFT computation. A free water molecule was primarily modeled as illustrated in Fig. 4d, with a bond angle of 104.5° and O-H bond length ($L_{OH}$) of 0.96 Å. The water inter-action study was then conducted on the theoretical structures of $ZIS_T$ and $gZIS_T$, specifically on the basal plane where the $S_v$ is located. It is found that the water molecule experiences a weak physisorption and mild activation (Fig. 4e and Supplementary Fig. 8a–c) upon interacting with the Zn atom of pristine $ZIS_T$ structure. On the other hand, the unsaturated Zn atom of $gZIS_T$ possesses a lower electron density as discussed in Fig. 3d, and thus a relatively higher partial positive charge ($\delta^+$) from the asymmetric distribution of electrons. From the nature of higher electronegativity of O as compared to H atom, the O atom holds a partial negative charge ($\delta^-$) in the water molecule and tends to attract to any oppositely charged atom. Consequently, the $\delta^+$-Zn in $gZIS_T$ could drive a more intense water interaction with the $\delta^-$-O from water. Comparing the water interactivity on the basal plane of the structures, $gZIS_T$ is more competent in activating the water molecule than $ZIS_T$ (see Fig. 4e, f and Supplementary Figs. 8a–f) as demonstrated by the greater bond angle expansion and $L_{OH}$ elongation. The water adsorp-tion free energy $\left(\Delta E_{H_2O^*}\right)$ at the basal plane of $gZIS_T$ (−1.40 eV) is also found to be more negative than that of $ZIS_T$ (−0.97 eV), demonstrating a more favorable adsorption of water towards activated basal plane of $gZIS_T$ from thermochemical perspective. Interestingly, the defect location of $S_v$ is able to accommodate water molecule adsorption (see Fig. 4g and Supplementary Figs. 8g–i). The unsaturated Zn atom near to $S_v$ strongly interacts with the adsorbed water molecule, leading to more sizable $L_{OH}$ lengthening and bond angle stretching to ease the O-H bond breaking. In conjunction with the exposure of more surface

active sites, gZIS could further elevate the photocatalytic efficiency by promoting water molecule interaction and propagating the cleavage of O-H bond for $H^+$ deprotonation to drive both the HER and OER forward.

## Electrochemical attributes and charge transfer properties

Transient photocurrent responses and electrochemical impedance spectroscopy (EIS) Nyquist analysis were conducted to assess the charge transfer dynamics of the samples. As depicted in Fig. 5a, the measured photocurrent intensity of gZIS is approximately twice as much as the pristine ZIS, contributing to a striking enhancement of photogenerated electron-hole pairs separation. EIS Nyquist plot is also provided in Fig. 5b to consolidate the charge transfer kinetics. The plot is fitted according to an equivalent Randle circuit consisting of inter-facial charge transfer resistance ($R_{CT}$), series resistance ($R_S$) and con-stant phase element (CPE) for the electrolyte-electrode interface (see inset of Fig. 5b and Supplementary Table 2). In detail, the arc diameter of the plot corresponds to the charge transfer impedance whereby the smaller the arc size of the semicircle, the lower the $R_{CT}$ value, the faster the charge transfer and separation of the photogenerated charge carriers[43–45]. As observed, gZIS displays a smaller semicircle arc accompanied by a reduction of $R_{CT}$, proving that the $S_v$-induced charge redistribution and asymmetric dipole moment facilitating the charge transfer within the framework of gZIS. Moreover, steady-state photoluminescence (PL) spectrums for both ZIS and gZIS were recorded in Fig. 5c. The pristine ZIS manifests a single prominent peak, emitting from the band-to-band transition across the band gap fol-lowed by the recombination of photogenerated charge carriers. Con-trarily, gZIS displays two distinctive peaks which are attributed to the intrinsic band-to-band radiative transition of excited electron from conduction band (CB) to empty state in valence band (VB), and the extrinsic sub-band from defect state introduced by $S_v$ to the ground state[46]. Not only that, the intensity of gZIS PL spectra is quenched due to the presence of $S_v$ acting as electron trap to restrain the electron-hole pairs recombination, and simultaneously expedite the charge

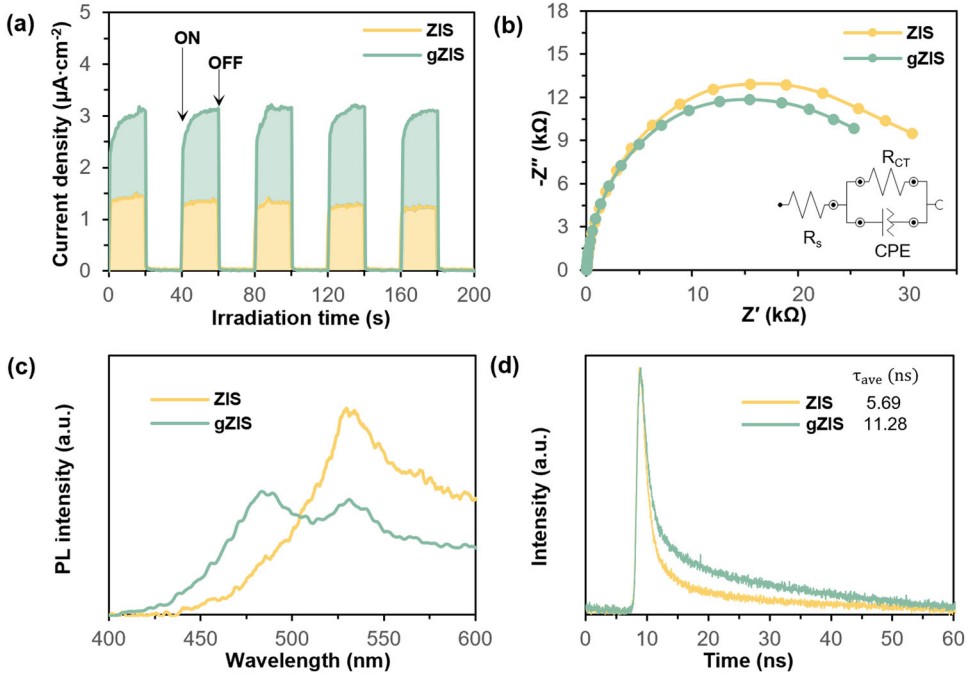

**Fig. 5 | Photoelectrochemical and charge transfer characteristics. a** Transient photocurrent responses, (**b**) EIS Nyquist plot with the equivalent Randle circuit, (**c**) steady-state PL emission spectra and (**d**) transient TRPL decay spectra of ZIS and gZIS.

transfer process for active participation in the photocatalytic reaction. It is generally accepted that the empirical lifetime of electron ($\tau'_e$) is approximated by the equation of $\tau'_e = \frac{1}{2\pi f_{max}}$, whereby $f_{max}$ corresponds to the frequency of the maximum Bode-phase peak[47]. Based on the reciprocal correlation and Bode-phase plot in Supplementary Fig. 9, the negative shift in $f_{max}$ of gZIS implies a prolonged electron lifetime as compared to that of ZIS, which is associated to the augmented charge carrier separation. Time-resolved PL (TRPL) was then carried out to verify the lifetime behavior of the samples. As depicted in Fig. 5d and Supplementary Table 3, both the fluorescence decay curves follow a bi-exponential decay model, whereby gZIS possesses longer average lifetime ($\tau_{ave}$) than pristine ZIS. Thus, it can be deduced that the charge separation efficiency of gZIS is significantly enhanced, allowing more photogenerated charge carriers to actively participate in reactions.

**Luminescent-electrochemical properties evaluation**

It is crucial to investigate the photo-absorption characteristics and the energy band positions of the samples in order to correlate to the performance of photocatalytic $H_2$ production. The optical absorption properties were examined by ultraviolet-visible (UV-Vis) diffuse reflectance spectra as shown in Fig. 6a. Pristine ZIS is found to have the ability to absorb the blue-to-green region of the visible light spectrum, reflecting a large portion of unutilized yellow-to-red visible light and appearing to be bright orangish-yellow as shown in the inset. The gZIS sample, however, experiences a blue shift of intrinsic absorption edge due to the size reduction as observed[14,48]. Astoundingly, gZIS exhibits an extended absorption tail up to the red visible and near infrared (NIR) region, where pristine ZIS displays nil responses beyond its absorption edge. This intriguing phenomenon may be attributed to the presence of $S_v$ within gZIS framework, introducing an additional defect state that can utilize both the high and low photon energy to drive two-step photoexcitation of electrons. In other words, gZIS has the tendency to capture the short and long wavelength of light, featuring its intrinsic band gap and the extrinsic defect sub-band. This unique light-absorbing property is analogous to that of natural chlorophyll, which absorbs the blue and red region of the light spectrum, giving leaves their characteristic green color. Therefore, gZIS also

exists in a green hue as opposed to the customary yellow color. The optical band gaps ($E_g$) of the samples are obtained from the Kubelka-Munk function vs. the incident photon energy plots as elucidated in Fig. 6b[49]. It is found that both ZIS and gZIS possess visible-light-active $E_g$ of 2.31 and 2.63 eV, respectively. In addition, the energy level of the band tail in gZIS, or commonly known as Urbach's tail, is calculated using the Urbach equation to locate the position of defect state (see details in Supplementary Fig. 10a)[50]. The inverse slop of the linearized Urbach equation dictates an Urbach energy ($E_u$) of 0.34 eV from the CB, representing the location of defect sub-band as a shallow trap state near to the CB of gZIS. Besides, transition energy ($E_t$) evaluation could also serve as a technique to represent the intraband state, whereby the numerical value of $E_t$ could be obtained via extrapolation of the Tauc plot to the x-axis[51]. Concordant with the $E_u$ evaluation, $E_t$ suggests the exact same level of defect state which is situated well below the CB of gZIS (see Supplementary Fig. 10b). In short, the presence of $S_v$-induced defect sub-band is verified by the PL spectra, $E_u$ calculation and $E_t$ computation (see Supplementary Fig. 11), signifying the capability of gZIS to harvest the high energy photon to promote photo-excitation of electron from the ground state, and concomitantly provide an alternate lower energy excitation route to facilitate secondary excitation of electron by utilizing the long wavelength electromagnetic radiation.

With the aim of acquiring fundamental understanding of the electrochemical properties of the samples, Mott-Schottky (MS) measurements were performed as presented in Fig. 6c. The positive slopes observed in the samples validate the n-type behaviors, as is usually reported[52]. Furthermore, it is unequivocally presented that the gradient of gZIS is lower than that of ZIS, indicating an elevated concentration of donor charge carrier ($N_D$) of gZIS based on the inverse-proportionality relationship between gradient and charge density. Specifically, the MS gradient and charge density are related following the equation of $N_D = \frac{2}{q\varepsilon\varepsilon_0} \cdot \frac{1}{gradient}$, whereby the governing constants q, $\varepsilon$, and $\varepsilon_0$ represent the elementary charge constant, material dielectric constant and vacuum permittivity, respectively[53]. Upon utilizing the aforementioned equation, it is intriguing to observe that the $N_D$ of gZIS ($8.23 \times 10^{21}$ cm$^{-3}$) exceeds that of ZIS ($6.94 \times 10^{21}$ cm$^{-3}$), attributed

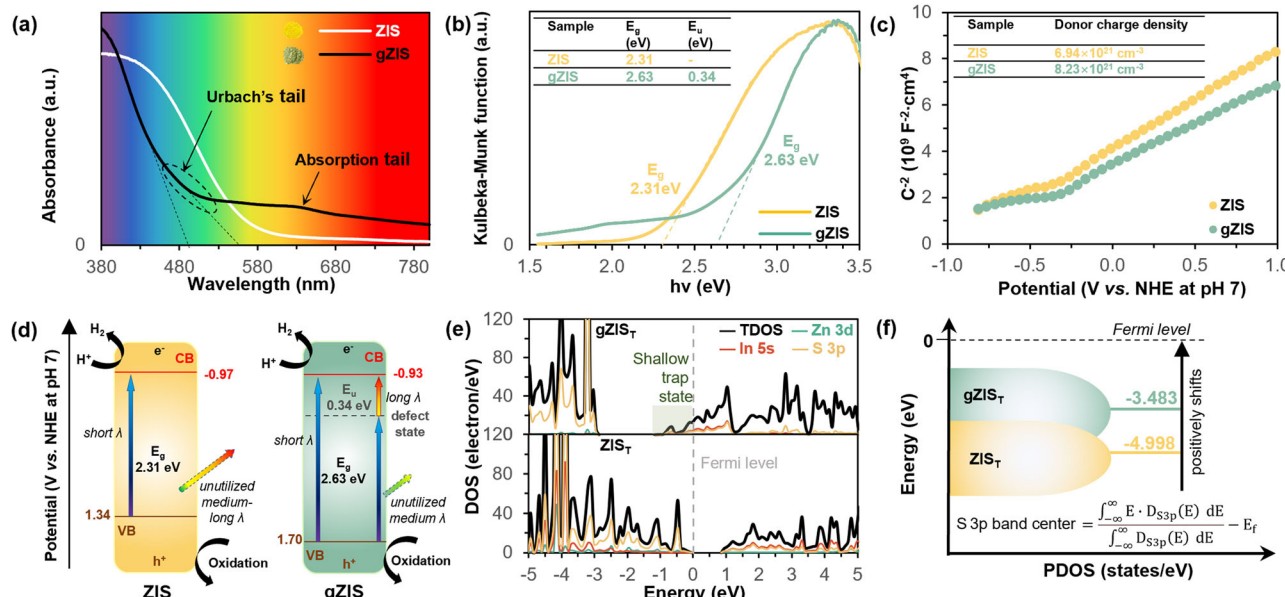

**Fig. 6 | Optoelectronic properties and band structure. a** UV-Vis diffuse reflectance spectra with inset showing the actual color of the samples, (**b**) KM function for band gap determination, and (**c**) MS plot for ZIS and gZIS. **d** Schematic of the electronic band structures of ZIS and gZIS with light absorption properties and photogeneration electron-holes pair formation mechanisms. **e** Theoretical calculated DOS and (**f**) respective $\varepsilon_p$ for ZIS$_T$ and gZIS$_T$.

to the favorable $S_v$-induced charge redistribution and electron delocalization. The mechanism is capable of generating a denser amount of charge carriers that can contribute in elevating photocatalytic performances. On top of that, the valence band edge ($E_{VB}$) of the samples was evaluated via ultraviolet photoelectron spectroscopy (UPS) analysis. As shown in Supplementary Fig. 12, ZIS and gZIS possess $E_{VB}$ of 1.34 and 1.70 V vs. NHE at pH 7, respectively. The conduction band edge ($E_{CB}$) of the samples can be further evaluated through the expression of $E_{CB} = E_{VB} - E_g$, in conjunction with the values obtained. Leveraging all the antecedently acquired outcomes, the proposed band structures of the samples are illustrated in Fig. 6d. As depicted, the pristine ZIS could utilize only the photon energy equal to or greater than its $E_g$ for photogeneration of electrons and holes to drive the respective photocatalytic reactions at the CB and VB. On the contrary, gZIS could exploit both the high and low energy light spectrum to drive the photocatalytic reaction, by primarily harvesting high-energy photons to initiate the photoexcitation of ground-state electrons directly to the CB for the photoreductive reaction. Concurrently, gZIS provides an alternative lower energy channel through absorption of medium-long wavelength photons to excite electrons from VB to CB via the defect state. Furthermore, the presence of the defect state will trap energy-lost electrons from the CB, thereby preventing them from returning to the ground state and recombining with photogenerated holes. The temporarily captured electrons in the defect state could undergo secondary photonic excitation back to the CB to drive the photoreduction reaction by absorbing low-energy light illumination, signifying the benefits of the defect state in utilizing solar energy to its full advantage to generate charge carriers. DFT calculations were performed to elucidate the electronic potential of $S_v$-induced defect state onto the structure. In agreement with the experimental findings, the density of state (DOS) computations in Fig. 6e unveils the presence of shallow trap state near to the CB of sulfur-vacant gZIS$_T$. The DOS profiles also demonstrate the enhancement of electron density around VB of gZIS$_T$, which implies a greater proportion of readily available ground state electrons to be photoexcited for photocatalytic HER. Expanding the calculation from the DOS, S 3p band center ($\varepsilon_p$) could be evaluated as shown in Fig. 6f, whereby $D_{S3p}(E)$ denotes the energy-dependent DOS projected onto the p orbitals of S element and $E_f$

dictates the Fermi level of the system (conventionally set to 0 eV). The theoretical $\varepsilon_p$ of gZIS$_T$ (−3.483 eV) is discovered to positively shift and approach the $E_f$ as compared to that of ZIS$_T$ (−4.998 eV). In accordance with the p-band center correlation, $\varepsilon_p$ governs the surface activity, *i.e.*, a smaller deviation of $\varepsilon_p$ from $E_f$ indicates a greater electron accumulation around the species with a stronger adsorption capability[54–56]. Moreover, the gZIS possesses a reduction in work function with the introduction of $S_v$, which is beneficial for surficial electron transfer (Supplementary Figs. 13 and 14). These empirical findings collectively suggest a higher charge accumulation around active S sites to facilitate H* adsorption and promote photoelectron transfer for augmented HER.

## Water splitting mechanisms and performance

To unravel a more comprehensive insight towards HER, H* adsorption Gibbs free energy ($\Delta G_{H^*}$) was evaluated (see details under Supplementary Fig. 15). According to the Sabatier principle, an ideal HER photocatalyst should compromise both the adsorption and desorption kinetic barriers in expediting electron transfer to the bonded H* as well as spontaneous release of generated H$_2$ from the surface, indicated by a thermoneutral value of $\Delta G_{H^*}$.[57,58] Analogizing the most active (110) facet of the structure, gZIS$_T$ endows a more favorable theoretical HER kinetics than pristine ZIS$_T$ as shown by the closer-to-zero $\Delta G_{H^*}$ in Fig. 7a, which favors the adsorptive reduction of H$^+$ to form H$_2$ through intimate electron transfer. Credited to the $S_v$-induced charge redistribution around S atoms, the inert basal plane is activated to be more propitious towards H* adsorption on the surface with a lower HER barrier. Interestingly, the unoccupied defect location ($S_v$) also allows an additional binding of H* for catalyzing the reductive generation of H$_2$ as shown in Supplementary Fig. 15d. In brief, the introduction of $S_v$ into the ZIS structure not only diminishes the adsorption-desorption barriers at the intrinsically active (110) site to better drive the HER, but also simultaneously triggers the catalytic activity of inert basal planes as to provide additional reactive HER centers. DFT calculations were extended to elucidate the effect of $S_v$ toward interfacial O$_2$ evolution reaction (OER) mechanisms (refer Supplementary Figs. 16 and 17 for details). As summarized in Fig. 7b, it could be observed that the basal (110) plane of pristine ZIS$_T$ does not

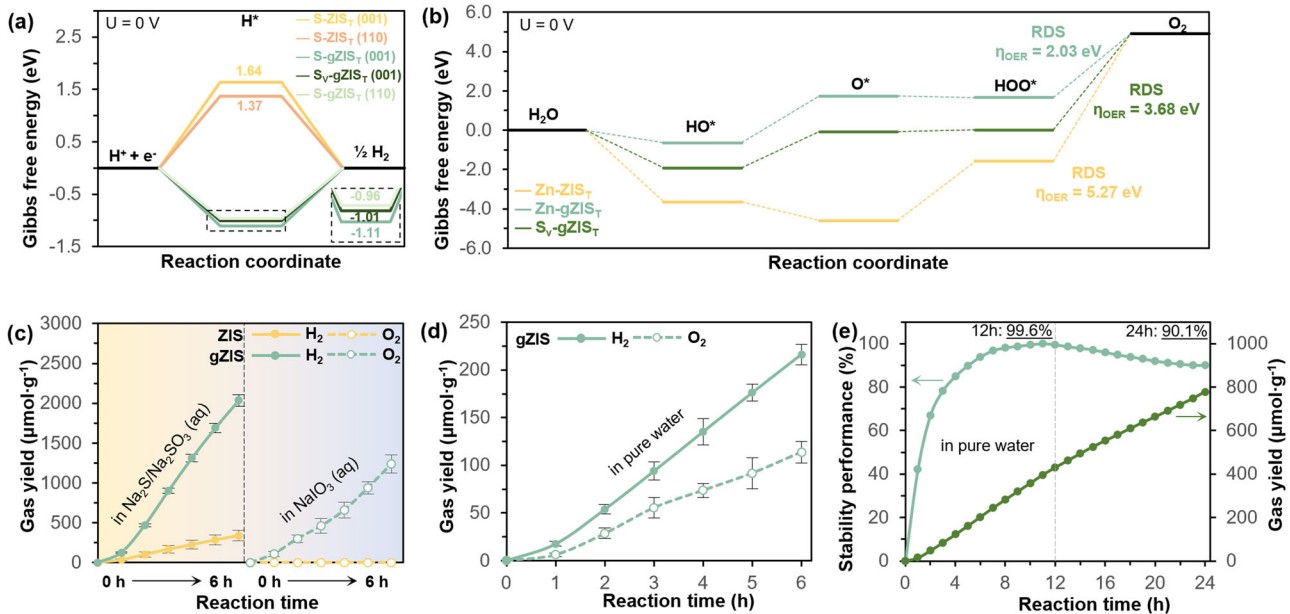

**Fig. 7 | Water splitting mechanism and performance.** Gibbs free energy maps for (**a**) HER and (**b**) OER for ZIS$_T$ and gZIS$_T$. **c** Photocatalytic HER and OER half-reaction under different sacrificial conditions. **d** Time-dependent solar-driven overall water splitting performance and (**e**) long-term photocatalytic stability performance of gZIS. Error bars represent the standard deviation from two independent runs.

favor OER with a large overpotential ($\eta_{OER}$) of 5.27 eV in the process of O$_2$ production from HOO*. Conversely, the unsaturated Zn atom of gZIS$_T$ is found to be more conducive towards OER with a generally diminished energy barrier. Despite the O$_2$ disengagement process persisting as the rate-determining step (RDS), the $\eta_{OER}$ is incredibly reduced to 2.03 eV which signifies the advantage of S$_v$ towards decreasing OER energy barriers, facilitating the H$_2$O oxidation and simultaneously escalating HER by providing more H$^+$ from the deprotonation of H$_2$O. Magnificently, the deprotonation process of H$_2$O on the unsaturated Zn to form HO* intermediate is barrierless for gZIS$_T$ (low negative value of −0.65 eV), representing a thermodynamically favorable process. Thus, the H$_2$O oxidation process is capable of competing with self-oxidation of sulfide from high oxidation potential of photogenerated holes, which in turn impeding photocorrosion of sulfide and catalyzing stable OER process[59,60]. Besides, the defect location induced by S$_v$ formerly found to be capable in activating H$_2$O molecule (see Fig. 4g) could also provide an alternative pathway for OER with an $\eta_{OER}$ (3.68 eV) lower than that of pristine ZIS$_T$. In order to scrutinize the catalytic performance of ZIS and gZIS, photocatalytic half-reactions (HER and OER) were primarily performed. Under Na$_2$S/Na$_2$SO$_3$ sacrificial conditions as shown in Fig. 7c, pristine ZIS presented a H$_2$ evolution of 338.33 μmol•g$^{-1}$ under six-hour continuous light radiation. Remarkably, gZIS exhibited distinctive performance enhancement with six-hour continuous H$_2$ evolution of 2036.41 μmol•g$^{-1}$, that is more than 6-fold performance than the pristine counterpart. A controlled experiment was performed to validate that H$_2$ generation was indeed driven by the reduction of H$^+$ by the photogenerated electrons in the CB (see details in Supplementary Fig. 18). Briefly, additional NaIO$_3$ was introduced into the system to scavenge electrons as to inhibit the photoreductive production of H$_2$. It was evidently reflected that nil H$_2$ could be observed under Na$_2$S/Na$_2$SO$_3$ + NaIO$_3$ sacrificial conditions, confirming the generation of H$_2$ was driven by photoelectrons in the CB of gZIS whereby the S$^{2-}$/SO$_3$$^{2-}$ is irreversibly oxidized by holes to form S$_2$O$_3$$^{2-}$/SO$_4$$^{2-}$ (ref. 61). Following that, pristine ZIS did not exhibit any O$_2$ evolution under NaIO$_3$ sacrificial conditions, whereas gZIS demonstrated a significant O$_2$ production of 1239.92 μmol•g$^{-1}$ under 6-h of irradiation. Extending from the capability of gZIS catalyzing both the HER and OER process,

solar-driven overall pure water splitting experiment was conducted without any sacrificial reagent. The gZIS employed the competence to drive photocatalytic pure water splitting with H$_2$ and O$_2$ yield of 36.04 and 18.96 μmol•g$^{-1}$•h$^{-1}$, respectively (close to a stoichiometric ratio of 2:1), while pristine ZIS did not deliver any appreciable yield. Additional controlled experiments were also performed in the dark and without photocatalyst to eliminate the potential false-positive observation from the background and photolysis of water. The controlled experiments did not present any H$_2$ evolution (Supplementary Fig. 19). Moreover, there is a consistency trend between the optical absorption property of gZIS with its respective AQY at different monochromatic wavelengths (Supplementary Fig. 20), indicating the H$_2$ is in fact generated via photon utilization of gZIS in solar-driven water splitting. The stability of gZIS in photocatalytic pure water splitting was also examined as shown in Fig. 7e. The gZIS retained 99.6% performance for a full daytime irradiation (12-hour equivalence) and still possessed more than 90% performance after one full solar day (24-h) reaction, with negligible changes in post-reaction characterizations (Supplementary Figs. 21–23). This fascinating observation suggested gZIS refraining from the serious photocorrosion issues which is commonly experienced by other sulfide-based catalysts. On one hand, this single-component cocatalyst-free gZIS displays a remarkable H$_2$ half-reaction AQY of 5.34% (420 nm) that is comparable to other noble-metal loaded and complex ZIS-based heterostructure systems as reported in Supplementary Table 4. On the other hand, the ability of this self-activated and stable gZIS without introduction of heteroatom nor noble metal to drive photocatalytic overall water splitting reaction with AQY (0.17%, 420 nm) and STH (0.002%), can be on par with the other modified and assisted sulfide-based photocatalysts (Supplementary Table 5).

In summary, a unique superhydrophilic green gZIS was successfully constructed via an in-situ solvothermal strategy. In-depth experimental investigations and theoretical computations conducted in this study systematically unraveled the fundamental insights on the critical roles of morphology transformation, surface modification, and vacancy engineering. The efficient photocatalytic water splitting activity of self-activated gZIS is attributed to the exclusive hollow hierarchical framework, exposing more intrinsically active facet

and activating the inert basal plane, as well as the presence of super-hydrophilic surface enhancing water interaction. These intriguing occurrences allow gZIS to maximize the utilization of surface areas. Additionally, the presence of $S_v$ within the structure propagates significant charge redistribution and induces asymmetric dipole moment, which consequently boosts the charge transfer, reduces the surface HER and OER kinetic barriers. The existence of defect state in the electronic band structure of gZIS further expands the optical absorption properties and mediates photoexcitation of electrons via alternative two-step process. Besides exhibiting more than 6-fold enhancement in photocatalytic half-reaction of $H_2$ production than conventional yellow pristine ZIS, this gZIS could also catalyze solar-driven overall water splitting reaction with high stability, and performance comparable to other complex sulfide-based photocatalysts. This self-activated high activity single-component noble-metal-free gZIS contains high value of exploration and could open up a brand-new design opportunity. It is believed that this could encourage the generation of novel ideas to conceive and devise a highly efficient gZIS-based photocatalyst to sustainably drive large-scale green $H_2$ production for achieving a carbon-neutral future.

## Methods

### Materials

Analytical grade reagents were used directly without any purification. Zinc chloride ($ZnCl_2$, Merck, ≥98%), indium (III) chloride tetrahydrate ($InCl_3 \cdot 4H_2O$, Sigma Aldrich, ≥97%), thioacetamide ($C_2H_5NS$, Nacalai Tesque, ≥99%), ethylene glycol ($C_2H_6O_2$, Sigma Aldrich, ≥99%), ethanol ($C_2H_5OH$, Fisher Scientific, ≥96%). Deionized water (DI water, resistivity ≥18 MΩ•cm) used in this experiment was obtained from Millipore Milli-Q water purification system.

### Synthesis of pristine and hollow $ZnIn_2S_4$ microsphere

Pristine $ZnIn_2S_4$ (ZIS) was synthesized via one-step hydrothermal method, where stoichiometric ratio of 0.5 mmol $ZnCl_2$, 1.0 mmol $InCl_3 \cdot 4H_2O$ and 2.0 mmol $C_2H_5NS$ were dissolved homogeneously in 30 mL DI water. The solution was transferred into a Teflon vessel held in a stainless-steel autoclave maintained at 160 °C for 12 h. After cooling to room temperature, the solution was subjected to thorough washing with ethanol and DI water to completely remove any unreacted precursor. Yellow ZIS powder was obtained upon overnight freeze drying. A similar process was used to obtain green hollow $ZnIn_2S_4$ (gZIS) powder by replacing DI water with ethylene glycol (EG) in a solvothermal synthesis process.

### Materials characterizations

The surface morphology and elemental composition of the samples were analyzed by field emission scanning electron microscopy (FESEM) using the Hitachi SU8010 microscope equipped with an energy-dispersive X-ray (EDX). Transmission electron microscopy (TEM) and high-resolution TEM (HRTEM) imaging were taken using the JEOL, JEM-2100 F microscope. Atomic-resolution spherical aberration-corrected bright field scanning TEM (BF-STEM) imaging with fast Fourier transformation (FFT) was obtained from the Hitachi HD-2700. The crystallographic properties and information of the samples were obtained via X-ray diffraction (XRD) analysis by utilizing the Bruker D8 Discovery X-ray diffractometer with an Ni-filtered Cu Kα radiation. X-ray photoelectron spectroscopy (XPS) analysis of surface chemical states were obtained using the Thermo Fisher Scientific Nexsa G2 XPS with monochromatic Al-Kα (hν = 1486.6 eV) X-ray source. The binding energies were referenced to adventitious carbon signal (C 1 s peak) at 284.6 eV prior to peak deconvolution. Ultraviolet photoelectron spectroscopy (UPS) analysis was performed using the Thermo Fisher Scientific Nexsa G2 surface analysis system by using vacuum UV radiation for induction of

photoelectric effects. The photon emission possessed an energy of 21.22 eV through He I excitation. The contact potential differences of the materials were obtained through Kelvin probe force microscopy (KPFM) using Bruker Multimode 8 atomic force microscope (AFM) electric mode. The sample powders were evenly spray-coated on fluorine-doped tin oxide (FTO) glass and mounted onto AFM sample stage with silver paste to ensure uninterrupted electrical connection. The surface area information of the samples was obtained from the multipoint Brunauer-Emmett-Teller (BET) $N_2$ adsorption-desorption isotherm at 77 K using the Micrometrics ASAP 2020. The samples were subjected to degassing at 150 °C for 8 h to remove any adsorbed species prior to the analysis. Surface wettability test and water contact angle measurement were conducted using the Ramè-hart Co Model 250 goniometer. In this context, 3 µL droplets of DI water was adapted as working medium to drop onto sample-coated FTO glass slide to perform contact angle analysis with triplicate measurement data collected. The electron paramagnetic resonance (EPR) measurements were performed at room temperature using a spectrometer (JEOL, JES-FA200). Ultraviolet-visible (UV-Vis) diffused reflectance spectra of the samples were obtained from the Agilent Cary 100 UV-Vis spectrophotometer equipped with an integrated sphere and $BaSO_4$ as reflectance standard. The optical band gap was obtained from the Kulbeka-Munk relationship. Steady-state photoluminescence (PL) spectra was acquired from the Perkin Elmer LS55 fluorescent spectrometer. Time-resolved PL (TRPL) spectra was recorded using the DeltaPro Fluorescence lifetime system (Horiba Scientific) with an excitation wavelength of 317 nm.

### Photoelectrochemical analysis

Photoelectrochemical (PEC) measurements including transient photocurrent response, electrochemical impedance spectroscopy (EIS) and Mott-Schottky plots were conducted using Metrohm Autolab electrochemical workstation. A conventional three-electrode PEC setup was adapted with 0.5 M $Na_2SO_4$ (pH = 7) as the electrolyte solution. Platinum (Pt) served as the counter electrode whereby Ag/AgCl saturated with 3.0 M KCl was utilized as the reference electrode. The working electrode was prepared by uniformly coating the sample onto FTO glass substrate with an active square area of 1 cm by 1 cm. The working electrode was illuminated by a 500 W Xe arc lamp with a fixed sample-to-lamp distance of 10 cm during the PEC analysis. A potential of +0.2 V was applied for the transient photocurrent and the working electrode was exposed to the light source at an intermittent light on-off rate of 20 s interval. Subsequently, EIS measurements were performed across a frequency range from 10 mHz to 100 kHz, with an equivalent Randle circuit was fitted according to the obtained Nyquist plot. Lastly, the Mott-Schottky plots were measured in the range from −1.0 to 0.8 V vs. Ag/AgCl with a potential step of 50 mV at a frequency of 1 kHz. For standardization, normal hydrogen electrode (NHE) scale at pH 7 was adapted as in $E_{NHE} = E_{Ag/AgCl} + 0.059pH + 0.1967V$[61].

### Photocatalytic hydrogen and oxygen evolution half-reaction

In photocatalytic $H_2$ evolution half-reaction, 30 mg of photocatalyst was dispersed homogenously in a 60 mL aqueous solution containing 0.35 M $Na_2S/Na_2SO_3$. The solution was then transferred into a Pyrex top-irradiated vessel with quartz window. The outlet of the vessel was connected to the Agilent 7820 A gas chromatography (Ar carrier gas) for gas measurement at hourly sampling interval. Prior to photocatalytic performance analysis, the system was purged with a high flowrate of $N_2$ gas for at least half an hour. The reactor was illuminated using 500 W Xe arc lamp with AM1.5 filter (*c.a.* 1 Sun illumination) during the reaction. Photocatalytic $O_2$ evolution half-reaction was carried out under the same conditions except that 0.1 M $NaIO_3$ was adopted as the sacrificial reagent.

## Photocatalytic overall water-splitting reaction

Photocatalytic overall water splitting reaction was carried out under the exact same condition as photocatalytic hydrogen and oxygen evolution half-reaction as described, except that the solution consists of pure DI water without the presence of any sacrificial reagent. Besides, the apparent quantum yield (AQY) was evaluated under different monochromatic light under various band pass filters (355, 420 and 500 nm) following the equation:

$$AQY(\%) = \frac{2N_{H_2}}{N_p} \times 100\% = \frac{2r_{H_2} N_A hc}{ISt\lambda} \times 100\% \quad (1)$$

in which $N_{H_2}$ = total number of $H_2$ molecules evolved, $N_p$ = total number of incident photons, $r_{H_2}$ = amount of $H_2$ molecule generated at time t (in mol), $N_A$ = Avogadro constant, h = Planck constant, c = speed of light, I = light intensity, S = irradiation area and $\lambda$ = wavelength of monochromatic light. Following that, solar-to-hydrogen (STH) conversion efficiency was determined at 1 Sun illumination with AM 1.5 filter in concordance to the equation:

$$STH(\%) = \frac{R_{H_2} \Delta G_r}{P_{Sun} S} \times 100\% \quad (2)$$

whereby $R_{H_2}$ = rate of $H_2$ evolution (in mol•s$^{-1}$), $\Delta G_r$ = Gibbs free energy change of water splitting reaction and $P_{sun}$ = energy flux of the incident ray.

## Computational details

Density functional theory (DFT) computations were conducted using Vienna Ab initio simulation package (VASP)[62]. Exchange-correlation potential was described using generalized gradient approximation (GGA) with Perdew-Burke-Ernzerhof (PBE) parameterization[63]. 500 eV energy cut-off, $1 \times 10^{-5}$ energy convergence and 0.01 eV•Å$^{-1}$ force converged were adapted as plane wave basis settings in this study. The Monkhorst-Pack k-point mesh was set at $3 \times 3 \times 1$. A theoretical $2 \times 2$ bilayer unit cell of pristine ZIS (ZIS$_T$) and sulfur-vacant gZIS (gZIS$_T$) structures were modeled by removing different intrinsic S atoms from the framework. All the atoms were allowed to relax with an additional 15 Å vacuum layer added perpendicular to the surface to eliminate any potential periodic image interaction. Hybrid functional Heyd-Scuseria-Ernzerhof (HSE06) was employed in the density of state and band edge evaluations[64]. For $H_2$ adsorption and water interaction study, 4-by-4 supercell was adopted as the substrate to neglect any interactions of adsorbates with adjacent unit cells. Grimmer's DFT-D3 method was accounted as additional van der Waals (vdW) correction for higher accuracy in computing the interatomic forces, stress tensor and potential energy[65]. Reaction Gibbs free energy ($\Delta G_r$) calculations for the HER and OER processes were evaluated utilizing:

$$\Delta G_r = \Delta E_r + (\Delta ZPE - T\Delta S)_r - eU \quad (3)$$

wherein $\Delta E_r$ is the reaction adsorption energy, $\Delta ZPE$ is the zero-point energy correction factor, $T\Delta S$ is the temperature dependent entropy contribution and eU is the external bias accounting the elementary proton-coupled transfer step.

## Data availability

The data supporting the findings of this study are available within the article and its Supplementary Information. The source data is available from the corresponding author upon reasonable request.

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

## Acknowledgements

This research project was funded by the Malaysia Research University Network (MRUN) from the Ministry of Higher Education Malaysia (Grant No. 304/PJKIMIA/656501/K145) and MUM-ASEAN Research Grant Scheme (Ref. No. ASE-000010) from Monash University Malaysia. This work was also supported by the High Impact Research Support Fund (HIRSF) (Ref. No. REU00354) and Advanced Computing Platform (APC) from Monash University Malaysia. We thank Hong Yuan Tok from Hi-Tech Instruments Sdn. Bhd. for the spherical aberration-corrected BF-STEM measurements.

## Author contributions

W.-K.C. carried out the sample synthesis, characterization, and theoretical calculations as well as wrote the paper. B.-J.N., Y.J.L., L.-L.T., L.K.P., and S.-P.C. discussed and validated the experimental and theoretical results. J.L. performed EPR analysis and validated the results. B.-J.N., L.-L.T., A.R.M., and S.-P.C. supervised the project. All authors contributed to the overall scientific interpretation and revised this paper.

## Competing interests

The authors declare no competing interests.
