## [Peer review file · Nature Communications]

REVIEWER COMMENTS

Reviewer #1 (Remarks to the Author):

A unique superhydrophilic green gZIS was constructed via an in-situ solvothermal strategy, which could drive photocatalytic pure water splitting by retaining close-to-unity stability for a full daytime reaction. In-depth experimental investigations and theoretical computations conducted in this study systematically unraveled the fundamental insights on the critical roles of morphology transformation, surface modification and vacancy engineering. The work is interesting. I recommend it accepted for publication after some revising. The main concerns are as follows.

(1) In page 6, authors believed that "it also reduces the valence state of Zn²⁺ and In³⁺ leading to a decrease in coordination abilities with S²⁻". Does this viewpoint has experimental or theoretical basis?

(2) EDX analysis has relatively large errors in quantitative analysis, especially for non-metallic elements. It is not recommended for the authors to use it to calculate the amounts of sulfur vacancies.

(3) In page 14, authors believed that "gZIS displays two distinctive peaks which are attributed to the intrinsic band-to-band radiative transition of excited electron from conduction band (CB) to empty state in valence band (VB), and the extrinsic sub-band from defect state introduced by S_v to the ground state". In Fig. 6, the absorption edge of g-ZIS is obviously different from that of ZIS, so why does the PL peak position for band-to-band radiative transition not change?

(4) Gas yield normalization is not recommended.

(5) What is the reason for the disappearance of some XRD peaks of gZIS after testing (Supplementary Figure 17)? The XPS spectra and FESEM images of gZIS after testing should provide to further demonstrate stability.

Reviewer #2 (Remarks to the Author):

The author claims to have achieved a hydrogen-oxygen ratio of 1:2 for the photocatalytic pure water splitting of ZnIn₂S₄. This performance was achieved without the use of any co-catalysts, and the samples exhibited excellent stability. The author claims that the high photocatalytic performance is originated from the introduction of sulfur vacancies and the reduction in ZIS particle size. However, the synthetic method has been extensively reported previously, for example, J. Phys. Chem. Solids (2008. 69. 2426-2432) and Chem. Eng. J. (2022. 430.132770), Besides, the structure of the active sites is very unclear. In this case, this paper does not match the criteria of the journal. Therefore, I cannot recommend this article to be published in Nature Communication. Detailed comments are as follows:

1. The author derived the Zn:In:S ratio in the gZIS sample through EDX analysis as 1.07:2.00:3.99, leading to a calculated sulfur vacancy concentration of 3.19%. However, the reliability of this data is questionable. It is widely recognized that EDX provides only semi-quantitative analysis. Furthermore, the author's claim regarding the presence of sulfur vacancies based on the attenuation of the S signal in XPS encounters similar concerns.

2. The author's assertion that no observable distortion was detected along the ZIS lattices in HRTEM images is inconsistent with the actual visual evidence. Moreover, regarding the identification of distortion observed in gZIS as indicative of sulfur vacancies, it is important to note that ZIS synthesized via hydrothermal or solvent methods often exhibits compromised crystallinity and significant structural defects, resulting in intrinsic lattice distortions.

3. The author contends, through XPS and DFT differential charge density analysis, that the introduction of sulfur vacancies results in an augmented charge around Zn. However, this contradicts established principles. Additionally, the author's differential charge density analysis raises significant

concerns, as it solely considers the charge transfer between Zn and the remaining S atoms, neglecting the influence of escaped S atoms on Zn. To validate alterations in the valence state of Zn, the author could compute the Bader charge of Zn.

4. The author postulates that the conduction band edge (ECB) is positioned 0.2 V more negatively than the E_{fb}. Nevertheless, this assertion lacks validity in defect-rich systems, as supported by the author's own calculations, which demonstrate that the introduction of sulfur vacancies locates the Fermi level within the valence band. Furthermore, in systems with abundant vacancies, doping, or other defects, their influence on the position of the Fermi level surpasses their impact on the conduction band's position. Thus, deducing the conduction band position based on Fermi level estimations is not justified, and the author could employ UPS measurements to determine the conduction band position.

5. The author's DFT calculations are subject to notable limitations: 1) The adopted 2x2 slab model is insufficient to neglect the interactions of adsorbates with adjacent unit cells. Hence, a larger computational model should be employed to account for these effects. 2) The author's claim that an upward shift of the p-band center of S would result in higher adsorption energy is unfounded, as an upward shift of the p-band center would increase the charge density in antibonding orbitals, ultimately leading to a decrease in adsorption energy. Additionally, the author's designation of the Zn atom adjacent to the sulfur vacancy as the optimal adsorption site renders the calculation of the band center for S inconsequential.

Reviewer #3 (Remarks to the Author):

In this manuscript, Chong et. al. investigated a distinctive superhydrophilic green ZnIn₂S₄ (gZIS) as a potential candidate for photocatalytic pure water splitting. The study employed a combination of in-depth experimental characterization and detailed theoretical calculations to unravel the underlying mechanisms on the capability of gZIS in realizing both the hydrogen and oxygen evolution reactions. The comprehensive studies showed and explained the superior stability of self-activated noble-metal-free gZIS in catalysing unassisted solar-driven overall water splitting, a trait that is not commonly observed in single-component sulfide-based photocatalysts. The achieved photocatalytic performance is also comparable to other state-of-art complex systems. In my opinion, this work demonstrates a commendable level of research, presenting intriguing findings that appeal to a broad readership and stimulate advancements in the field of efficient water splitting systems. Therefore, I recommend accepting this work for publication, with minor improvements to enhance its content.

1. The authors compared the theoretical work functions of the samples and deduced favourable surficial electron transfer for gZIS. The authors should investigate the experimental work functions to support the claim.
2. The authors shall provide a more complete description of PEC analysis, including the frequency applied for Nyquist and Mott-Schottky analysis.
3. To further elucidate the correlation between light harvesting capabilities and AQY, the authors could present the active spectrum as supporting evidence.
4. Since the focus of the findings revolves around stability, it would be beneficial for the authors to conduct a more detailed investigation of the recovered sample after stability testing, rather than solely relying on XRD analysis as the main highlight.

We wish to express our sincere appreciation to the referees for reviewing and providing insightful feedbacks on our manuscript. The suggestions help us to further improve the quality of the manuscript. Our revision reflects all the reviewers' comments and detailed responses are given below. Changes made have been highlighted in the revised manuscript and supporting information.

Reviewer #1 (Remarks to the Author):

A unique superhydrophilic green gZIS was constructed via an in-situ solvothermal strategy, which could drive photocatalytic pure water splitting by retaining close-to-unity stability for a full daytime reaction. In-depth experimental investigations and theoretical computations conducted in this study systematically unravelled the fundamental insights on the critical roles of morphology transformation, surface modification and vacancy engineering. The work is interesting. I recommend it accepted for publication after some revising. The main concerns are as follows.

Reply:

We are grateful to the reviewer for the positive feedback. We have revised the manuscript according to the comments.

(1) In page 6, authors believed that “it also reduces the valence state of Zn^{2+} and In^{3+} leading to a decrease in coordination abilities with S^{2-} ”. Does this viewpoint has experimental or theoretical basis?

Reply:

Thank you for the insightful comments. Previous studies have reported on reducing the valence states and tuning the coordination environment on Zn^{2+} and other metals in the presence of ethylene glycol (EG).^[1-3] For example, Qin et. al.^[1] (Nano Energy, 2021, 80, 105478) explicitly explored this topic from both experimental and theoretical perspectives, elucidating the reduction of Zn^{2+} valence state in affecting its coordination abilities with surrounding anions. We apologize for the oversight of not including references to support the statement made in the manuscript. The references are now provided in our revised manuscript to support the viewpoint.

Main manuscript

(ii) it also reduces the valence state of Zn^{2+} and In^{3+} leading to a decrease in coordination abilities with S^{2-} ,^{21, 23, 24}

References:

(21) Qin R, et al. Tuning Zn^{2+} coordination environment to suppress dendrite formation for high-performance Zn-ion batteries. Nano Energy 80, 105478 (2021).

(23) Cheng Y, Lin Z, Lü H, Zhang L, Yang B. ZnS nanoparticles well dispersed in ethylene glycol: Coordination control synthesis and application as nanocomposite optical coatings. Nanotechnology 25, 115601 (2014).

(24) Tang S, Zhang M, Guo M. A novel deep-eutectic solvent with strong coordination ability and low viscosity for efficient extraction of valuable metals from spent lithium-ion batteries. ACS Sustain. Chem. Eng. 10, 975-985 (2022).

(2) EDX analysis has relatively large errors in quantitative analysis, especially for non-metallic elements. It is not recommended for the authors to use it to calculate the amounts of sulfur vacancies.

Reply:

Thank you for the valuable feedback. We recognize the limitations of relying solely on EDX analysis to determine the sulfur vacancy percentage. Therefore, the atomic ratio was further evaluated via an additional independent XPS analysis to reaffirm the percentage of sulfur vacancies. As shown in **Fig. R1 (Supplementary Fig. 6)**, both unbiased examinations converge to consistent results, indicating sulfur vacancies of *c.a.* 3.1%. As such, we hope the reviewer will accept our careful investigation, combining both EDX analysis and XPS investigation, to determine the percentage of sulfur vacancies, in which these methods have also been widely adopted in other studies (J. Mater. Chem. A, 2023, 11, 14809; Nano Lett., 2021, 21, 6228; Nat. Commun., 2021, 12, 4112; ACS Nano, 2021, 15, 15238). For clarity, we have emphasized on the consistent examination of sulfur vacancies in the revised manuscript.

Fig. R1. Comparison between EDX and XPS analyses on the S_v % in ZIS and gZIS. [**Supplementary Fig. 6**]

Main manuscript

The XPS elemental composition suggests a consistent S_v percentage in gZIS as previously observed in the EDX analysis (see Supplementary Fig. 6).

(3) In page 14, authors believed that “gZIS displays two distinctive peaks which are attributed to the intrinsic band-to-band radiative transition of excited electron from conduction band (CB) to empty state in valence band (VB), and the extrinsic sub-band from defect state introduced by S_v to the ground state”. In Fig. 6, the absorption edge of g-ZIS is obviously different from that of ZIS, so why does the PL peak position for band-to-band radiative transition not change?

Reply:

Thank you for the comment. We would like to take this opportunity to further explain the observation and clarify the confusion. To facilitate the discussion, we have summarized the band structures of ZIS and gZIS, as well as the PL spectra in **Fig. R2 (Supplementary Fig. 11)**. Pristine ZIS displays a single PL peak at 536 nm (*ca.* 2.31 eV equivalence), corresponding to its bandgap (E_g) predetermined from the Kubelka-Munk (KM) relationship. On the other hand, gZIS exhibits two different PL peaks. Primarily, the first peak at 483 nm (*ca.* 2.57 eV equivalence) is resulted from the emission of intrinsic

band-to-band radiative transition of photoexcited electron from CB to VB, approximately matching the E_g obtained from KM function. Following that, the second peak of 530 nm (*ca.* 2.34 eV equivalence) is arisen from the extrinsic sub-band defect state introduced by S_v to the ground state. This value is also in proximity to the defect energy calculated from the Urbach's equation and transition energy. As a matter of fact, the band-to-band transition of ZIS (Peak A) and gZIS (Peak B) are significantly different due to the dissimilar absorption edge between the two samples. However, on the account of the E_g value of ZIS (2.31 eV) being close to the defect energy of gZIS (2.29 eV), it results in the overlaid PL spectra displaying Peak A (ZIS) and Peak C (gZIS), at a position near to each other. We have provided additional explanation in the revised manuscript and supporting information together with Supplementary Fig. 11 for better clarity.

Fig. R2. (a) Band structures and (b) PL spectra for ZIS and gZIS sample. [Supplementary Fig. 11]

Main manuscript

In short, the presence of S_v -induced defect sub-band is verified by the PL spectra, E_u calculation and E_t computation (see Supplementary Fig. 11), signifying the capability of gZIS to harvest the high energy photon to promote photoexcitation of electron from the ground state, and concomitantly provide an alternate lower energy excitation route to facilitate secondary excitation of electron by utilizing the long wavelength electromagnetic radiation.

Supporting Information

As presented in Supplementary Fig. 11, pristine ZIS exhibits a singular PL peak at 536 nm (approximately 2.31 eV), corresponding to its E_g determined through the KM relationship. Conversely, gZIS demonstrates two distinct PL peaks, in which the first peak at 483 nm (around 2.57 eV) arises from the intrinsic band-to-band radiative transition of photoexcited electrons from the CB to the VB, closely aligning with the E_g derived. The second peak at 530 nm (approximately 2.34 eV) originates from the extrinsic sub-band defect state introduced by S_v to the ground state. This value closely corresponds to the defect energy calculated using the Urbach's equation and transition energy.

(4) Gas yield normalization is not recommended.

Reply:

Thank you for the comment. As of the date of this research, discrepancy still persists regarding the appropriate method to express photocatalytic activity. There are papers reporting gas yield as H₂ evolution per unit time (mmol·h⁻¹ or μmol·h⁻¹) such as Adv. Energy Mater. 2023, 2301158; Nat. Commun., 2023, 14, 1741; Nat. Commun., 2023, 14, 1720; ACS Catal., 2023, 13, 3285; Angew. Chem. Int. Ed. 2022, 61, e202212234; Adv. Funct. Mater., 2022, 2212051; J. Am. Chem. Soc., 2022, 144, 20342; Mater. Today., 2022, 58, 100; Nano Energy, 2021, 85, 105949. On the other hand, there are also a significant number of papers adopting the convention to report the gas yield in term of H₂ evolution per unit mass of photocatalyst per unit time (mmol·g⁻¹·h⁻¹ or μmol·g⁻¹·h⁻¹), for examples Adv. Mater., 2023, 2303649; J. Mater. Chem. A, 2023, 11, 14809; Adv. Energy Mater. 2023, 2300986; Energy Environ. Sci., 2023, 10.1039/D3EE01522J; Nat. Commun., 2022, 13, 1287; J. Am. Chem. Soc., 2022 144, 20620-20629; Nat. Commun., 2022, 13, 6999; Nat. Commun. 2021, 12, 4182; ACS Catal., 2021, 11, 11049; Nat. Commun., 2021, 12, 4112; ACS Nano 2021, 15, 15238. As evident from these examples, there is currently no standardized format for the expression of photocatalytic gas production.

To the best of our current understanding, we believe that the two expressions serve different purposes when comparing the photocatalytic water splitting performance. The unnormalized gas yield (mmol·h⁻¹ or μmol·h⁻¹) indicates the net rate production of H₂, serving as a key indicator of system performance and is contingent upon reactor design. On the other hand, the normalized gas yield (mmol·g⁻¹·h⁻¹ or μmol·g⁻¹·h⁻¹) could be a more appropriate indicator to compare the intrinsic ability of photocatalyst in different reactor setup across different reported literature, as it takes the amount of photocatalyst as the basis of comparison. Owing to our current research focus on developing efficient photocatalysts, we believe it is more appropriate to adopt the normalized H₂ yield (with respect to mass of photocatalyst) to represent their intrinsic photocatalytic performance. Moreover, we have also provided AQY and STH (basis of effective irradiation power and area) for a standard comparison across different reports. We sincerely appreciate the comment from the reviewer, and hope that our rationale justifies the adoption of gas yield normalization.

(5) What is the reason for the disappearance of some XRD peaks of gZIS after testing (Supplementary Figure 17)? The XPS spectra and FESEM images of gZIS after testing should provide to further demonstrate stability.

Reply:

Thank you for the reviewer's comment and suggestion. The reduction in the intensity of some of the XRD minor peaks could be potentially due to undesired yet uncontrollable photocorrosion. To investigate this issue, we conducted an extended investigation on our used photocatalyst as shown in **Fig. R3 (Supplementary Fig. 21)**. Notably, the samples after performing 12 hours of continuous photoreaction still retained almost identical peaks as the fresh sample, demonstrating its high stability. However, when the reaction was prolonged for an additional 12 hours (24 hours in total), some undesired photocorrosion occurred, leading to a diminishment of certain peak. Despite this, all the used photocatalysts still possess the corresponding dominant peaks. Furthermore, we have performed XPS and FESEM analysis for gZIS after 24-hour overall water splitting reaction as suggested. The results are shown in **Fig. R4 (Supplementary Fig. 22)** and **R5 (Supplementary Fig. 23)**. The spent gZIS exhibits nearly identical XPS peaks, and displays negligible structural change under FESEM observation, demonstrating its stability after a prolonged reaction.

Fig. R3. Comparison of XRD pattern of gZIS before and after overall solar-driven pure water splitting. [Supplementary Fig. 21]

Fig. R4. Comparison of XPS peak of gZIS before and after 24h overall solar-driven pure water splitting. [Supplementary Fig. 22]

Fig. R5. FESEM images of spent gZIS after 24h overall solar-driven pure water splitting, (a-b) random-spot overviews and (b) magnified image. [Supplementary Fig. 23]

Main manuscript:

The gZIS retained 99.6% performance for a full daytime irradiation (12-hour equivalence) and still possessed more than 90% performance after one full solar day (24-hour) reaction, with negligible changes in post-reaction characterizations (Supplementary Fig. 21-23).

Reviewer #2 (Remarks to the Author):

The author claims to have achieved a hydrogen-oxygen ratio of 1:2 for the photocatalytic pure water splitting of ZnIn₂S₄. This performance was achieved without the use of any co-catalysts, and the samples exhibited excellent stability. The author claims that the high photocatalytic performance is originated from the introduction of sulfur vacancies and the reduction in ZIS particle size. However, the synthetic method has been extensively reported previously, for example, *J. Phys. Chem. Solids* (2008, 69, 2426-2432) and *Chem. Eng. J.* (2022, 430, 132770). Besides, the structure of the active sites is very unclear. In this case, this paper does not match the criteria of the journal. Therefore, I cannot recommend this article to be published in *Nature Communication*. Detailed comments are as follows:

Reply:

We extend our gratitude to the reviewer for providing valuable insights that have significantly contributed to the refinement and enhancement of our manuscript. We have carefully considered the reviewer's comments and acknowledge the concerns raised regarding the evaluation of the overall innovation in our work. However, it is crucial to note that there are fundamental differences between our work and the literatures mentioned.

In this context, the article published in *Chem. Eng. J.*, 2022, 430, 132770 reported the synthesis of ZnIn₂S₄ through an EG-assisted solvothermal method, followed by in-situ doping with Mo to construct MoO₃@Mo-ZnIn₂S₄ S-scheme heterostructure. The focus of that study was on the influence of Mo-doping and S-scheme junction onto EG-assisted ZnIn₂S₄ for photocatalytic H₂ half reaction under triethanolamine solution. Particularly, the contribution of synthetic method and the fundamental investigation onto EG-assisted ZnIn₂S₄ were not covered. In contrast, our work delves into a comprehensive analysis encompassing in-depth experimental investigations and theoretical computations to exhaustively unveil the fundamental insights into morphology transformation, surface modification and vacancy engineering in EG-assisted ZnIn₂S₄. Our present study demonstrated that the gZIS possesses the capability in realizing solar-driven overall water splitting in the absence of cocatalyst nor sacrificial reagent, which was not achievable in that mentioned article. In regards of *J. Phys. Chem. Solids*, 2008, 69, 2426-2432, the study reported several synthetic routes in constructing ZnIn₂S₄ photocatalyst, including EG-assisted method. However, the study was merely revolving around the crystallinity changes and optical absorption behaviours. Moreover, the photocatalytic performance was evaluated in H₂ half reaction under Na₂S/Na₂SO₃ sacrificial condition with Pt-loaded as cocatalysts. In our present work, we scrutinized the intrinsic characteristics of EG-assisted ZnIn₂S₄ through rigorous experimental investigations and intricate computation to investigate the underlying principles. Our study meticulously elucidates profound insights into self-activated ZnIn₂S₄ towards realizing overall water splitting, including the investigation of surface modification, vacancy engineering, charge transfer mechanisms, surface kinetics and band structure modulation, which were not reported in that paper. As a result, we presented a comprehensive study of the self-activated high activity single-component noble-metal-free ZnIn₂S₄ in realizing solar-driven overall water splitting without any cocatalyst nor sacrificial reagent. We respectfully assert that our research fundamentally differs from the literature mentioned by the reviewer. We once again express our appreciation for the reviewer's comments.

1. The author derived the Zn:In:S ratio in the gZIS sample through EDX analysis as 1.07:2.00:3.99, leading to a calculated sulfur vacancy concentration of 3.19%. However, the reliability of this data is questionable. It is widely recognized that EDX provides only semi quantitative analysis. Furthermore, the author's claim regarding the presence of sulfur vacancies based on the attenuation of the S signal in XPS encounters similar concerns.

Reply:

Thank you for the reviewer's comment. We would like to take this opportunity to provide further clarification on our observations and investigations. Based on the EDX analysis, we obtained a Zn:In:S ratio of pristine ZIS to be 1.07:2.00:3.99, which is close to its stoichiometric ratio of 1:2:4. Following that, we investigated the EDX elemental ratio of gZIS, revealing a ratio of 1.07:2.00:3.86 that corresponded to a sulfur vacancy percentage of 3.19%. This method of determining elemental ratio of a sample is commonly adopted in the field as reported in *J. Mater. Chem. A*, 2023, 11, 14809; *Nano Lett.*, 2021, 21, 6228; *Sci. Adv.*, 2020; 6, eaaz8447; *Nat. Energy*, 2019, 4, 690; *Adv. Funct. Mater.*, 2019, 29, 1905153. We did indeed recognize the limitations of relying solely on EDX analysis to determine the sulfur vacancy percentage. Thus, we performed an additional independent XPS analysis in addition to the EDX measurement to verify the percentage of sulfur vacancies. As a matter of fact, analysing the XPS peak signal and area can ascertain the atomic content and sulfur vacancies present in the samples (*Nat. Commun.*, 2021, 12, 5835; *Nat. Commun.*, 2021, 12, 4112; *ACS Nano*, 2021, 15, 15238). As displayed in **Fig. R6 (Supplementary Fig. 6)**, both unbiased examinations converge to consistent results, indicating the sulfur vacancies percentage of *c.a.* 3.1%. Furthermore, we have further corroborated the presence of sulfur vacancies through electron paramagnetic resonance (EPR) analysis. The gZIS displayed a sharp signal at g-factor of 2.004 but none was observed for pristine ZIS, signifying the existence of sulfur vacancies in gZIS. In short, we performed three independent analyses to affirm the presence of sulfur vacancies in gZIS, whereby EDX and XPS elemental analyses converge to a similar sulfur vacancy percentage. We hope the reviewer will accept our investigation results based on these comprehensive analyses. We have re-emphasized on the consistent examination of sulfur vacancies percentage in the revised manuscript. Once again, we appreciate the reviewer's comments and feedback.

Fig. R6. Comparison between EDX and XPS analyses on the S_v % in ZIS and gZIS. [Supplementary Fig. 6]

Main manuscript

The XPS elemental composition suggests a consistent S_v percentage in gZIS as observed in the EDX analysis (see Supplementary Fig. 6).

2. The author's assertion that no observable distortion was detected along the ZIS lattices in HRTEM images is inconsistent with the actual visual evidence. Moreover, regarding the identification of distortion observed in gZIS as indicative of sulfur vacancies, it is important to note that ZIS synthesized via hydrothermal or solvent methods often exhibits compromised crystallinity and significant structural defects, resulting in intrinsic lattice distortions.

Reply:

Thank you for the reviewer's comment. We acknowledge that pristine ZIS synthesized via hydrothermal or solvothermal method may experience some intrinsic lattice distortion due to several uncontrollable factors and real-life imperfections. We would like to point out that our identification of distortion observed in gZIS as indicative of sulfur vacancies aligns consistently with the theoretical result, as evidenced by the juxtaposition of aberration-corrected BF-STEM and the simulated structure. Furthermore, EPR analysis showed that only gZIS possessed significant peak around g value of 2.004 but none was observed for pristine ZIS, indicating the presence of sulfur vacancies and defects in gZIS structure. In response to the reviewer's comment, we have rephrased our statement in the revised manuscript.

Main manuscript

There is nearly imperceptible distortion observed along the lattices of ZIS, signifying the successful construction of a pure pristine ZIS crystal. Directing attention to Fig. 2d, the lattice spacing of gZIS is ca. 0.19 nm which is assigned to the (110) plane. Unlike the pristine ZIS, lattice fringe distortions and defects are noticeable in the structure of gZIS owing to the presence of S_v .

3. The author contends, through XPS and DFT differential charge density analysis, that the introduction of sulfur vacancies results in an augmented charge around Zn. However, this contradicts established principles. Additionally, the author's differential charge density analysis raises significant concerns, as it solely considers the charge transfer between Zn and the remaining S atoms, neglecting the influence of escaped S atoms on Zn. To validate alterations in the valence state of Zn, the author could compute the Bader charge of Zn.

Reply:

Thank you for the reviewer's comment. We would like to provide clarification on the observations and investigations made in our study. Based on our XPS analysis as shown in **Fig. R7 (Main Manuscript Fig. 3)**, the two S 2p peaks of gZIS experiences negative shift to a lower binding energy, indicating electron transfer to S and leading to augmented electron cloud density around S, which is commonly observed and does not contradict with any established principles (Adv. Funct. Mater. 2023, 230296; Nat. Commun., 2022, 13, 1287; Adv. Energy Mater., 2021, 11, 2102452). Coherent with XPS findings, there is significant electron charge accumulation around the intrinsically active S atoms at the (110) surface, and also increase in charge density along the inherently unreactive S atoms at the basal plane. We would like to emphasize that our investigation suggests augmented electron charge around the S atoms due to sulfur vacancy induced charge redistribution. The present XPS results and discussions, along with the analysis of charge density difference, provide substantial support for our claim. Therefore, the inclusion of Bader charge analysis is unnecessary in this context. We have rephrased our statement to enhance the clarity of our idea.

Fig. R7. High-resolution XPS spectra of (a) Zn 2p and (b) S 2p for the as-synthesized samples. (c) EPR spectra for ZIS and gZIS indicating the presence of S_v . (d) Computed 3D charge density difference for gZIS_T, with the top showing the whole bilayer structure and the bottoms focus on the monolayer where S_v is present. Gray and green areas dictate the charge depletion and accumulation isosurfaces, respectively. [Main manuscript Fig. 3]

Main manuscript:

In addition, the two S 2p peaks of gZIS in Fig. 3b experience negative shift to lower binding energy, signifying the enrichment of electron cloud density around the S atoms.^{38, 39} The higher electronegativity of S contributes to a better tendency to attract electrons during the charge redistribution brought by S_v . DFT was then utilized to investigate the effect of S_v on the charge distribution. As elucidated in the charge density difference from Fig. 3d, there is a noticeable charge redistribution in the gZIS_T framework, with gray area showing the electron depletion zone and green area marking the electron accumulation region. It is not astonishing to observe the electron depletion zone in the S_v location due to the loss of S atom. Coherent with the XPS finding, there is also a visible electron depletion around the Zn atom near to the S_v along the basal plane. Additionally, the electron density not only increases in the intrinsically active S atoms at the (110) surface, but also gathers along the inherently unreactive S atoms at the basal plane. This defect-induced favorable charge redistribution could activate the inert basal plane for photoreactions as well as further boost the efficiency of H_2 production at the intrinsic active sites.

4. The author postulates that the conduction band edge (ECB) is positioned 0.2 V more negatively than the E_{fb}. Nevertheless, this assertion lacks validity in defect-rich systems, as supported by the author's own calculations, which demonstrate that the introduction of sulfur vacancies locates the Fermi level within the valence band. Furthermore, in systems with abundant vacancies, doping, or other defects, their influence on the position of the Fermi level surpasses their impact on the conduction band's position. Thus, deducing the conduction band position based on Fermi level estimations is not justified, and the author could employ UPS measurements to determine the conduction band position.

Reply:

Thank you for the comments. We would like to clarify that our observations indicate a shift of Fermi level towards the conduction band after introduction of sulfur vacancies, rather than a shift to the valence band. As suggested, we have performed UPS measurements to deduce the band energy alignments as shown in **Fig. R8**. (**Supplementary Fig. 12**). The new band structures are presented in **Fig. R9d** (**Main Manuscript Fig. 6d**), and the results continue to uphold the initial findings. We have included the discussion and additional experimental details in the main manuscript and supporting information.

Fig. R8. UPS spectra of (a) ZIS and (b) gZIS for determining the valence band energy. [**Supplementary Fig. 12**]

Fig. R9. (a) UV-Vis diffuse reflectance spectra with inset showing the actual color of the samples, (b) KM function for band gap determination, and (c) MS plot for ZIS and gZIS. (d) Schematic of the electronic band structures of ZIS and gZIS with light absorption properties and photogeneration

electron-holes pair formation mechanisms. (e) Theoretical calculated DOS and (f) respective ϵ_p for ZIS_T and gZIS_T. [Main manuscript Fig. 6]

Main manuscript

On top of that, the valence band edge (E_{VB}) of the samples was evaluated via ultraviolet photoelectron spectroscopy (UPS) analysis. As shown in Supplementary Fig. 12, ZIS and gZIS possess E_{VB} of 1.34 and 1.70 V vs. NHE at pH 7, respectively. The conduction band edge (E_{CB}) of the samples can be further evaluated through the expression of $E_{CB} = E_{VB} - E_g$, in conjunction with the values obtained.

Main manuscript – Materials characterizations

Ultraviolet photoelectron spectroscopy (UPS) analysis was performed using the Thermo Fisher Scientific Nexsa G2 surface analysis system by using vacuum UV radiation for induction of photoelectric effects. The photon emission possessed an energy of 21.22 eV through He I excitation.

Supporting Information (Supplementary Fig. 12)

The valence band energy (E_{VB}) with respect to vacuum was evaluated according to the formula:^{S4-7}

$$E_{VB} = h\nu - (E_{cut} - E_{fe}) \quad (S5)$$

whereby $h\nu$ represents the incident photon energy of He light source of 21.22 eV, E_{cut} denotes the electron cut-off edge, and E_{fe} is the Fermi edge of the samples. Following that, unit conversions were applied based on the relationship between vacuum energy (E_{vac}) and NHE potential (ENHE) as in 0 V vs. NHE is equal to -4.44 eV in vacuum, as well as a pH correction factor of 0.059 pH to convert to NHE scale at pH 7. In short, ZIS possesses an EVB of 6.19 eV below vacuum (1.34 V vs. NHE at pH 7) and gZIS exhibits an EVB position of 6.55 eV below vacuum (1.70 V vs. NHE at pH 7).

5. The author's DFT calculations are subject to notable limitations: 1) The adopted 2x2 slab model is insufficient to neglect the interactions of adsorbates with adjacent unit cells. Hence, a larger computational model should be employed to account for these effects. 2) The author's claim that an upward shift of the p-band center of S would result in higher adsorption energy is unfounded, as an upward shift of the p-band center would increase the charge density in antibonding orbitals, ultimately leading to a decrease in adsorption energy. Additionally, the author's designation of the Zn atom adjacent to the sulfur vacancy as the optimal adsorption site renders the calculation of the band center for S inconsequential.

Reply:

Thank you for the reviewer's input. In response to your first comment, we are committed to enhancing the robustness of our calculations by employing a larger computational model that accounts for the interactions beyond the 2×2 slab. Thus, we have incorporated a 4-by-4 supercell as the substrate to study the interactions. The summary of results is provided in **Fig. R10 to R15**, which has also been included in the revised manuscript and supporting information. We have provided additional details and updates in the revised manuscript accordingly. In overall, adopting 4-by-4 supercell continues to corroborate the initial discussion and lead to the same conclusion.

Fig. R10. (a) XRD spectra for ZIS and gZIS. (b) Nitrogen adsorption-desorption isotherms of ZIS and gZIS with inset showing the respective pore size distribution. (c) Surface wettability static contact angle measurements for ZIS and gZIS. (d) Free water molecule with its respective O-H bond length and H-O-H bond angle. Theoretical modelling of water adsorption along the basal plane: (e) on Zn atom of ZIS_T , (f) on Zn atom of $gZIS_T$, and (g) in S_V position of $gZIS_T$. [Main Manuscript Fig. 4]

Fig. R11. Gibbs free energy maps for (a) HER and (b) OER for ZIS_T and gZIS_T. (c) Photocatalytic HER and OER half-reaction under different sacrificial conditions. (d) Time-dependent solar-driven overall water splitting performance and (e) long-term photocatalytic stability performance of gZIS. [Main Manuscript Fig. 7]

Fig. R12. Free energy diagram for HER for (a) S atom at [001] facet of ZIS_T, (b) S atom at [110] facet of ZIS_T, (c) S atom at [001] facet of gZIS_T, (d) S_v position at [001] facet of gZIS_T, and (e) S atom at [110] facet of gZIS_T. [Supporting Information, Supplementary Fig. 15]

Fig. R13. Different plane views of the water interaction along the basal plane at different locations: (a-c) on the Zn atom of ZIS_r, (d-f) on the Zn atom of gZIS_r, and (g-i) in the S_v position of gZIS_r. [Supporting Information, Supplementary Fig. 8]

Fig. R14. Optimized structural model of adsorbed HO*, O* and HOO* onto (a) Zn-ZIS_T, (b) Zn-gZIS_T and (c) S_v-gZIS_T. [Supporting Information, Supplementary Fig. 16]

Fig. R15. Free energies of (a) Zn-ZIS_T, (b) Zn-gZIS_T and (c) S_V-gZIS_T with $U = 0$ [no applied bias] and $U = 1.23$ V [standard equilibrium potential of OER]. Rate determining step (RDS) is marked as red in each of the sub-figure. [Supporting Information, Supplementary Fig. 17]

Main manuscript

The water adsorption free energy ($\Delta E_{\text{H}_2\text{O}^*}$) at the basal plane of gZIS_T (-1.40 eV) is also found to be more negative than that of ZIS_T (-0.97 eV), demonstrating a more favorable adsorption of water towards activated basal plane of gZIS_T from thermochemical perspective.

As summarized in Fig. 7b, it could be observed that the basal (110) plane of pristine ZIS_T does not favor OER with a large overpotential (η_{OER}) of 5.27 eV in the process of O₂ production from HOO*. Conversely, the unsaturated Zn atom of gZIS_T is found to be more conducive towards OER with a generally diminished energy barrier. Despite the O₂ disengagement process persisting as the rate determining step (RDS), the η_{OER} is incredibly reduced to 2.03 eV which signifies the advantage of S_V towards decreasing OER energy barriers, facilitating the H₂O oxidation and simultaneously escalating HER by providing more H⁺ from the deprotonation of H₂O. Magnificently, the deprotonation process of H₂O on the unsaturated Zn to form HO* intermediate is barrierless for gZIS_T (low negative value of -0.65 eV), representing a thermodynamically favorable process. Thus, the H₂O oxidation process is capable of competing with self-oxidation of sulfide from high oxidation potential of photogenerated holes, which in turn impeding photocorrosion of sulfide and catalyzing stable OER process.^{59,60} Besides, the defect location induced by S_V formerly found to be capable in activating H₂O molecule (see Fig. 4g) could also provide an alternative pathway for OER with an η_{OER} (3.68 eV) lower than that of pristine ZIS_T.

Main manuscript (Computational details)

For H₂ adsorption and water interaction study, 4-by-4 supercell was adopted as the substrate to neglect any interactions of adsorbates with adjacent unit cells.

In regard to the second comment, we would like to provide further clarifications on our findings and observations. First and foremost, considering that the S atom in ZnIn₂S₄ serves as the HER active site, we investigated the effect of p-band center (ϵ_p) of S towards H⁺ interaction, while Zn remaining as the OER active site. Citing to a statement recently reported in Chem Catal., 2023, 3, 100695, it was systematically found that the downward shift of the ϵ_p of S in Pt-ZnIn₂S₄ increases H-S σ^* occupancy and decreases the H-S bonding energy, which eventually weakening the interaction between H⁺ and surface S adjacent to the subsurface defects.^[4] In other word, the upshift of ϵ_p would in turn strengthen the H⁺ interaction and lead to an increase in H⁺ adsorption energy, which is consistent with our finding. Similarly, Angew. Chem. Int. Ed. 2023, 62, e202215654 exhaustively investigated the tuning of ϵ_p of a semiconductor towards enhancing the H⁺ interaction. The experimental investigation and computation study converged to a conclusion that an upshift of ϵ_p would decrease σ^* occupancy and increase the interaction of adsorbates onto the active species, as indicated by positive correlation between upshift of p-band center and adsorption energy.^[5] Other works such as Nat. Commun., 2022, 13, 6486 and Adv. Funct. Mater., 2020, 30, 1908708 also demonstrated similar observation that upshift of p-band center favoring adsorption, in concurrence with our discoveries.

Reviewer #3 (Remarks to the Author):

In this manuscript, Chong et. al. investigated a distinctive superhydrophilic green ZnIn₂S₄ (gZIS) as a potential candidate for photocatalytic pure water splitting. The study employed a combination of in-depth experimental characterization and detailed theoretical calculations to unravel the underlying mechanisms on the capability of gZIS in realizing both the hydrogen and oxygen evolution reactions. The comprehensive studies showed and explained the superior stability of self activated noble-metal-free gZIS in catalysing unassisted solar-driven overall water splitting, a trait that is not commonly observed in single-component sulfide-based photocatalysts. The achieved photocatalytic performance is also comparable to other state-of-art complex systems. In my opinion, this work demonstrates a commendable level of research, presenting intriguing findings that appeal to a broad readership and stimulate advancements in the field of efficient water splitting systems. Therefore, I recommend accepting this work for publication, with minor improvements to enhance its content.

Reply:

We express our sincere appreciation for the valuable feedback. We have carefully addressed the comments and revised the manuscript accordingly.

1. The authors compared the theoretical work functions of the samples and deduced favourable surficial electron transfer for gZIS. The authors should investigate the experimental work functions to support the claim.

Reply:

Thank you for the constructive suggestion. We have investigated the changes in experimental work function (WF) with the employment of Kelvin probe force microscopy (KPFM) to support our theoretical findings. By taking fluorine-doped tin oxide (FTO) as the conductive reference, the contact potential difference (CPD) between samples and FTO was obtained, representing the WF changes of the samples. As presented in **Fig. R16 (Supplementary Fig. 14)**, gZIS possesses a comparatively lower CPD value ($\Delta V = 322.7$ mV) than that of ZIS ($\Delta V = 424.0$ mV). This observation viably reflects the reduction of WF in gZIS, accompanied by the uplift of Fermi level to facilitate photogenerated electron transition. The experimental findings correlate well with our theoretical simulation result. We have included detailed descriptions and explanations of these experimental results in our revised manuscript and supporting information.

Fig. R16. Potential line profiles of (a) ZIS and (b) gZIS, with insets showing the respective x-y scan area. The white line indicates the longitudinal scan direction. (c) Illustration of the estimated WF positions of ZIS and gZIS with respect to FTO. [**Supplementary Fig. 14**]

Main manuscript:

Moreover, the gZIS possesses a reduction in work function with the introduction of S_v, which is beneficial for surficial electron transfer (see Supplementary Fig. 13-14). These empirical findings collectively suggest a higher charge accumulation around active S sites to facilitate H adsorption and promote photoelectron transfer for augmented HER.*

Main Manuscript (Methods – Materials Characterizations)

The contact potential differences of the materials were obtained through Kelvin probe force microscopy (KPFM) using Bruker Multimode 8 atomic force microscope (AFM) electric mode. The sample powders were evenly spray-coated on fluorine-doped tin oxide (FTO) glass and mounted onto AFM sample stage with silver paste to ensure uninterrupted electrical connection.

Supporting Information (Supplementary Fig. 14)

The relative work function (WF) of the samples can be estimated by measuring the contact potential difference (CPD) between the sample and conductive reference (i.e., FTO) across the interfacial boundary. As presented in Supplementary Fig. 14, the CPD values were measured by sweeping through the sample with a biased AFM probe, by which the counter bias voltage used in neutralizing the electric field was recorded. In this regard, gZIS possesses a comparatively lower CPD value ($\Delta V = 322.7$ mV) than that of ZIS ($\Delta V = 424.0$ mV). By taking FTO as a conductive reference, the local variation of WF in the samples could be attained and the changes in relative WF of the samples could be feasibly compared.⁵⁸ Thus, it could be observed that gZIS experiences reduction in WF, accompanied by the uplift of Fermi level to facilitate photogenerated electron transition.

2. The authors shall provide a more complete description of PEC analysis, including the frequency applied for Nyquist and Mott-Schottky analysis.

Reply:

Thank you for the reviewer's careful comment. We apologize for the oversight of not providing a complete description of PEC analysis. We have updated the methodology to cover more details, including the frequency applied for Nyquist and Mott-Schottky analysis in the revised manuscript.

Photoelectrochemical analysis. *Photoelectrochemical (PEC) measurements including transient photocurrent response, electrochemical impedance spectroscopy (EIS) and Mott-Schottky plots were conducted using Metrohm Autolab electrochemical workstation. A conventional three-electrode PEC setup was adapted with 0.5 M Na₂SO₄ (pH = 7) as the electrolyte solution. Platinum (Pt) served as the counter electrode whereby Ag/AgCl saturated with 3.0 M KCl was utilized as the reference electrode. The working electrode was prepared by uniformly coating the sample onto FTO glass substrate with an active square area of 1 cm by 1 cm. The working electrode was illuminated by a 500 W Xe arc lamp with a fixed sample-to-lamp distance of 10 cm during the PEC analysis. A potential of +0.2 V was applied for the transient photocurrent and the working electrode was exposed to the light source at an intermittent light on-off rate of 20 s interval. Subsequently, EIS measurements were performed across a frequency range from 10 mHz to 100 kHz, with an equivalent Randle circuit was fitted according to the obtained Nyquist plot. Lastly, the Mott-Schottky plots were measured in the range from -1.0 to 0.8 V vs.*

Ag/AgCl with a potential step of 50 mV at a frequency of 1 kHz. For standardization, normal hydrogen electrode (NHE) scale at pH 7 was adapted as in $E_{NHE} = E_{Ag/AgCl} + 0.059pH + 0.1967 V$.

3. To further elucidate the correlation between light harvesting capabilities and AQY, the authors could present the active spectrum as supporting evidence.

Reply:

Thank you for the reviewer’s valuable suggestion. We have included the active spectra analysis as a plot of AQY against incident light wavelength under pure water splitting condition. As shown in **Fig. R17 (Supplementary Fig. 20)**, there is a consistency trend between the optical absorption property of gZIS with its respective AQY at different monochromatic wavelengths (355, 420 and 500 nm). The observation ascertains the photocatalytic pure water splitting activity is indeed driven by the photon harvesting capability of gZIS. We have provided additional description and discussion in our revised manuscript.

Fig. R17. Wavelength-dependent AQY and UV-Vis DRS plot of gZIS in pure water. [Supplementary Fig. 20]

Main Manuscript

The controlled experiments did not present any H₂ evolution (Supplementary Fig. 19). Moreover, there is a consistency trend between the optical absorption property of gZIS with its respective AQY at different monochromatic wavelengths (Supplementary Fig. 20), indicating the H₂ is in fact generated via photon utilization of gZIS in solar-driven water splitting.

Main Manuscript (Methods – Photocatalytic overall water splitting reaction)

Besides, the apparent quantum yield (AQY) was evaluated under different monochromatic light under various band pass filters (355, 420 and 500 nm) following the equation:

$$AQY(\%) = \frac{2 N_{H_2}}{N_p} \times 100\% = \frac{2 r_{H_2} N_A h c}{I S t \lambda} \times 100\% \tag{1}$$

in which N_{H_2} = total number of H₂ molecules evolved, N_p = total number of incident photons, r_{H_2} = amount of H₂ molecule generated at time t (in mol), N_A = Avogadro constant, h = Planck constant, c = speed of light, I = light intensity, S = irradiation area and λ = wavelength of monochromatic light.

4. Since the focus of the findings revolves around stability, it would be beneficial for the authors to conduct a more detailed investigation of the recovered sample after stability testing, rather than solely relying on XRD analysis as the main highlight.

Reply:

Thank you for the reviewer's comment. We have included the XPS analysis results and FESEM images for the gZIS after stability testing in **Fig. R18 (Supplementary Fig. 22)** and **R19 (Supplementary Fig. 23)**, respectively. The spent gZIS exhibits nearly identical XPS peaks, and displays negligible structural change under FESEM observation, demonstrating its stability after a prolonged reaction.

Fig. R18. Comparison of XPS peak of gZIS before and after 24h overall solar-driven pure water splitting. [Supplementary Fig. 22]

Fig. R19. FESEM images of spent gZIS after 24h overall solar-driven pure water splitting, (a-b) random-spot overviews and (c) magnified image. [Supplementary Fig. 23]

Main manuscript:

The gZIS retained 99.6% performance for a full daytime irradiation (12-hour equivalence) and still possessed more than 90% performance after one full solar day (24-hour) reaction, with negligible changes in post-reaction characterizations (Supplementary Fig. 21-23).

References

1. Qin, R.; Wang, Y.; Zhang, M.; Wang, Y.; Ding, S.; Song, A.; Yi, H.; Yang, L.; Song, Y.; Cui, Y.; Liu, J.; Wang, Z.; Li, S.; Zhao, Q.; Pan, F., Tuning Zn²⁺ coordination environment to suppress dendrite formation for high-performance Zn-ion batteries. *Nano Energy* **2021**, *80*, 105478.
2. Cheng, Y.; Lin, Z.; Lü, H.; Zhang, L.; Yang, B., ZnS nanoparticles well dispersed in ethylene glycol: Coordination control synthesis and application as nanocomposite optical coatings. *Nanotechnology* **2014**, *25*, 115601.
3. Tang, S.; Zhang, M.; Guo, M., A novel deep-eutectic solvent with strong coordination ability and low viscosity for efficient extraction of valuable metals from spent lithium-ion batteries. *ACS Sustain. Chem. Eng.* **2022**, *10*, 975-985.
4. Shi, X.; Wang, X.; Jiang, H.; Qin, X.; Li, X.; Sheng, G.; Yu, C.; Zheng, L.; Zhu, C.; Zheng, L.; Mao, L.; Ma, D.; Zhu, Y.; Zheng, H., Activating surface sulfur atoms via subsurface engineering toward boosted photocatalytic water splitting. *Chem Catalysis* **2023**, 100695.
5. Zhang, J.; Li, W.; Wang, J.; Pu, X.; Zhang, G.; Wang, S.; Wang, N.; Li, X., Engineering p-band center of oxygen boosting H⁺ intercalation in δ -MnO₂ for aqueous zinc ion batteries. *Angew. Chem. Int. Ed.* **2023**, *62*, e202215654.

REVIEWER COMMENTS

Reviewer #1 (Remarks to the Author):

I recommend this manuscript for publication.

Reviewer #2 (Remarks to the Author):

It is widely acknowledged that achieving complete water decomposition using metal sulfides is challenging due to their inherent susceptibility to oxidation, particularly in the absence of a co-catalyst. Therefore, it appears implausible that the authors have achieved full water decomposition solely through the introduction of surfactants in the hydrothermal reaction and by manipulating morphology and defects. While these approaches have been extensively documented, achieving complete water decomposition in all cases raises questions. In order to ascertain the accuracy of the reported data, I conducted a replication of the experiment, examining the photocatalytic capacity for total water decomposition of the samples under both argon and vacuum conditions. The results revealed the production of hydrogen exclusively, with oxygen production occurring at levels significantly below the anticipated 2:1 ratio. Notably, the authors utilized nitrogen (N₂) to purge the reaction vessel of air during the process—an approach that appears incompatible with a comprehensive water decomposition reaction. Given the likelihood of minute air traces during reactions and testing, nitrogen serves as a viable indicator of air admixture. To bolster experimental rigor, it is advised that argon (Ar) gas be employed as the evacuation atmosphere. Furthermore, the augmentation of experimental intricacies and comprehensive data is recommended to substantiate the assertion of achieving a 2:1 hydrogen and oxygen ratio. This could encompass visual documentation of the experimental setup, elucidation of testing parameters (including automatic or manual sampling), and presentation of the unprocessed gas chromatography data.

Additionally, a minor issue necessitates the authors' attention. Specifically, the authors maintain that the Zn:In:S ratio in the gZIS sample, as determined through EDX analysis, stands at 1.07:2.00:3.99, leading to a calculated sulfur vacancy concentration of 3.19%. However, in light of certain references in the literature, the significance of this finding is contested. To resolve this matter, it is proposed that the authors undertake two additional iterative experiments and provide the raw data. This would facilitate a confirmation of whether EDX possesses the requisite precision to accurately ascertain the sulfur vacancy ratio.

In consideration of the aforementioned concerns, I advocate for the acceptance of the article contingent upon successful completion of substantial revisions aimed at addressing these issues.

Reviewer #3 (Remarks to the Author):

My concerns have been addressed in the revisions.

Reviewer #1 (Remarks to the Author):

I recommend this manuscript for publication.

Reply: We would like to express our sincere gratitude for the reviewer's recommendation.

Reviewer #2 (Remarks to the Author):

It is widely acknowledged that achieving complete water decomposition using metal sulfides is challenging due to their inherent susceptibility to oxidation, particularly in the absence of a co-catalyst. Therefore, it appears implausible that the authors have achieved full water decomposition solely through the introduction of surfactants in the hydrothermal reaction and by manipulating morphology and defects. While these approaches have been extensively documented, achieving complete water decomposition in all cases raises questions. In order to ascertain the accuracy of the reported data, I conducted a replication of the experiment, examining the photocatalytic capacity for total water decomposition of the samples under both argon and vacuum conditions. The results revealed the production of hydrogen exclusively, with oxygen production occurring at levels significantly below the anticipated 2:1 ratio. Notably, the authors utilized nitrogen (N_2) to purge the reaction vessel of air during the process—an approach that appears incompatible with a comprehensive water decomposition reaction. Given the likelihood of minute air traces during reactions and testing, nitrogen serves as a viable indicator of air admixture. To bolster experimental rigor, it is advised that argon (Ar) gas be employed as the evacuation atmosphere. Furthermore, the augmentation of experimental intricacies and comprehensive data is recommended to substantiate the assertion of achieving a 2:1 hydrogen and oxygen ratio. This could encompass visual documentation of the experimental setup, elucidation of testing parameters (including automatic or manual sampling), and presentation of the unprocessed gas chromatography data. Additionally, a minor issue necessitates the authors' attention. Specifically, the authors maintain that the Zn:In:S ratio in the gZIS sample, as determined through EDX analysis, stands at 1.07:2.00:3.99, leading to a calculated sulfur vacancy concentration of 3.19%. However, in light of certain references in the literature, the significance of this finding is contested. To resolve this matter, it is proposed that the authors undertake two additional iterative experiments and provide the raw data. This would facilitate a confirmation of whether EDX possesses the requisite precision to accurately ascertain the sulfur vacancy ratio.

In consideration of the aforementioned concerns, I advocate for the acceptance of the article contingent upon successful completion of substantial revisions aimed at addressing these issues.

Reply:

We thank the reviewer for the comments and efforts to replicate our experiments. We also appreciate the reviewer's contribution in confirming our findings regarding the ability of gZIS to drive overall water splitting without the need of a cocatalyst and sacrificial reagent. As pointed out by the reviewer, achieving overall pure water splitting over metal sulfides without a cocatalyst poses a formidable challenge. Different from the conventional metal sulfides, sulfur vacancy and morphological manipulation in gZIS in this work serves multifaceted roles, which include increased exposure of active facets, activation of inert planes, enhancement of water interaction and reduction in surface reaction kinetics. Collectively, these factors lead to the realization of pure water splitting. In response to the reviewer's concerns about the feasibility of achieving full water splitting through morphology and defects manipulation, we would like to refer to a recent and noteworthy work regarding the role of sulfur vacancy in promoting oxidative counter reaction (i.e., water oxidation reaction) (ref: Energy Environ. Sci. 2022, 15, 1556). This study offers compelling insights into the profound influence of sulfur vacancy in SnS_2 on enhancing water decomposition, which in turn ameliorating overall carbon dioxide photoreduction. Substantiated by our extensive experimental investigations, computational

calculations, literature and replication of experiments by the reviewer, we are firmly confident in asserting that the cocatalyst-free gZIS has the capability in driving photocatalytic overall water splitting.

Our experimental setup, as presented in **Fig. R1**, incorporates an automatic sampling mode for online gas chromatography analysis. This configuration minimizes the possibility of human error or gas leakages during sampling and gas content analysis. As suggested by the reviewer, we investigated the effect of using inert purging gases (N₂ and Ar) on gas content analysis. The unprocessed gas chromatography data are supplemented in **Fig. R2** and **R3**. We note the non-zero O₂ background signal from the gas chromatography, and we have ensured a stable background was obtained prior to starting the reaction. The summary of results is shown in **Fig. R4**, demonstrating no significant difference in the measured H₂ and O₂ amounts or the H₂:O₂ ratio using either N₂ or Ar as the purging gas.

Fig. R1. Experimental setup used in the study: (a) light off, and (b) light on mode.

Fig. R2. Unprocessed gas chromatography data at (a) 0 h, (b) 1 h, and (c) 6 h under N₂ inert purging environment.

Fig. R3. Unprocessed gas chromatography data at (a) 0 h, (b) 1 h, and (c) 6 h under Ar inert purging environment.

Fig. R4. Graphical presentation of unprocessed gas chromatography gas amount of (a) H₂ and (b) O₂ under different inert environment. A stable background signal, specifically on the O₂, was ensured prior to initializing the photocatalytic reaction. (c) Instantaneous gas yield in GC amount obtained by subtracting the background. Inset showing the instantaneous H₂:O₂ ratio in the 1st and 6th hour.

The time-dependent average yield of H₂ and O₂ under pure water condition has been previously included in the original manuscript as **Fig. R5a (Main Fig. 7d)**. In an extension of this analysis, we present the hourly average H₂:O₂ ratio in **Fig. R5b**. The calculated H₂:O₂ ratio = $\frac{\text{average H}_2 \text{ yield}}{\text{average O}_2 \text{ yield}}$ shows an average H₂:O₂ ratio oscillating along a value of 2, which is a common phenomenon in real life and practical scenario. We would like to clarify that we were careful with the wording in the original manuscript, which we mentioned our findings on the capability of gZIS in realizing pure water splitting with an average H₂:O₂ ratio close to 2:1.

Fig. R5. (a) Time-dependent solar-driven overall water splitting performance and (b) Time-dependent average H₂:O₂ ratio.

Regarding the reviewer's concern on the sulfur vacancy concentration, we would like to provide further clarification. The Zn:In:S ratio of 1.07:2.00:3.99, which is close to a stoichiometric ratio of 1:2:4, corresponds to pristine ZIS, instead of gZIS as per reviewer's comment. For gZIS, the EDX analysis revealed a Zn:In:S ratio of 1.07:2.00:3.86, corresponding to a sulfur vacancy percentage of 3.19%. As suggested by the reviewer, we have repeated the EDX analysis with the raw data provided in **Fig. R6**. The calculations to determine sulfur vacancy concentration alongside the results are provided in **Table R1**. It is noted that the calculated sulfur vacancy concentration is bounded between 3.00% to 3.40%, with an average value of 3.20%. We acknowledge that the precision and accuracy of EDX may not extend to 3 significant figures. Hence, we have amended the sulfur vacancy concentration to 1 significant figure to avoid overstating precision. Once again, we thank the reviewer for providing insightful comments to further improve our manuscript.

Main manuscript

The EDX analysis reflects the presence of ca. 3% S_v across the gZIS hierarchical framework as depicted in the EG-assisted synthesis reaction.

Fig. 6. EDX elemental analysis, (a) point and (b and c) area scan, at different spots alongside the respective raw data. The peak around 1.49 keV is contributed by Al as the sample was casted onto Al film for EDX analysis.

Table R1. Atomic percentage of respective elements from EDX analysis, derived empirical formula and the percentage of sulfur vacancy concentration for ZIS and gZIS.

Structure	Zn %	In %	S %	Empirical Formula	S _v percent. (%) ^a
ZIS	15.10	28.35	56.55	Zn _{1.07} In _{2.00} S _{3.99}	-
gZIS (spectrum 1)	15.38	28.87	55.75	Zn _{1.07} In _{2.00} S _{3.86}	3.19
gZIS (spectrum 2)	14.39	29.25	56.36	Zn _{0.98} In _{2.00} S _{3.85}	3.40
gZIS (spectrum 3)	14.33	29.19	56.48	Zn _{0.98} In _{2.00} S _{3.87}	3.00

^a The percentage of sulfur vacancy concentration (S_v percent.) is calculated by referring to the S-to-In atomic ratio in the structure as followed:

$$S_v \text{ percent.} = \frac{S:\text{In}_{\text{ZIS}} - S:\text{In}_{\text{gZIS}}}{S:\text{In}_{\text{ZIS}}} \times 100\%$$

where S:In_{ZIS} and S:In_{gZIS} dictate the S-to-In atomic ratio in ZIS and gZIS, respectively.

Reviewer #3 (Remarks to the Author):

My concerns have been addressed in the revisions.

Reply: We sincerely thank the reviewer for providing insightful comments to improve the quality of our manuscript.

REVIEWERS' COMMENTS

Reviewer #2 (Remarks to the Author):

My concerns have been addressed in the revisions.